# Localized Zeroth-Order Prompt Optimization

**Wenyang Hu** [* 1 2]  **Yao Shu** [* 3]  **Zongmin Yu** [1]  **Zhaoxuan Wu** [2 4]  **Xiaoqiang Lin** [1]  **Zhongxiang Dai** [5]
**See-Kiong Ng** [1 2]  **Bryan Kian Hsiang Low** [1]

## Abstract

The efficacy of large language models (LLMs) in understanding and generating natural language has aroused a wide interest in developing prompt-based methods to harness the power of black-box LLMs, especially through the lens of In-Context Learning. Existing methods usually prioritize a global optimization for finding the global optimum of prompts, which however will perform poorly in certain tasks. This thus motivates us to re-think the necessity of finding a global optimum in prompt optimization. To answer this, we conduct a thorough empirical study on prompt optimization and draw two major insights. Contrasting with the rarity of global optimum, local optima are usually prevalent and well-performed, which can be more worthwhile for efficient prompt optimization (**Insight I**). The choice of the input domain, covering both the generation and the representation of prompts, affects the identification of well-performing local optima (**Insight II**). Inspired by these insights, we propose a novel algorithm, namely *localized zeroth-order prompt optimization* (ZOPO), which incorporates a Neural Tangent Kernel-based derived Gaussian process into standard zeroth-order optimization for an efficient search of well-performing local optima in prompt optimization. Remarkably, ZOPO outperforms existing baselines in terms of both the optimization performance and the query efficiency, which we demonstrate through extensive experiments.

---

[1]Department of Computer Science, National University of Singapore [2]Institute of Data Science, National University of Singapore [3]Guangdong Lab of AI and Digital Economy (SZ) [4]Integrative Sciences and Engineering Programme, National University of Singapore [5]LIDS and EECS, Massachusetts Institute of Technology. Correspondence to: Yao Shu <shuyao@gml.ac.cn>.

*Proceedings of the 1st Workshop on In-Context Learning at the 41st International Conference on Machine Learning*, Vienna, Austria. 2024. Copyright 2024 by the author(s).

## 1. Introduction

In-context learning (ICL) has become an effective paradigm to help LLMs understand and generate appropriate responses when a few input-output pairs are provided to describe a specific task (Brown et al., 2020). However, the outputs of LLMs can be highly sensitive and biased to the given ICL exemplars (Min et al., 2022), leading to sub-optimal performance. On the other hand, in-context prompting also appears as an effective paradigm to instruct LLMs to generate desired outputs, where crafted prompts are added to the LLM's input (Ouyang et al., 2022). Particularly, prompting can further leverage LLMs' capability of understanding ICL exemplars and thus improve performances. Such an approach is of particular interest when users interact with state-of-the-art LLMs like ChatGPT (OpenAI, 2024a) and GPT-4 (OpenAI, 2023), which can only be accessed through black-box APIs (i.e., the interface of black-box LLMs only accepts discrete texts as input for querying). Therefore, optimizing in-context prompts becomes a critical effort in pursuing the optimal performance of black-box LLMs on downstream tasks.

Although human knowledge may subjectively guide prompt designs (Mishra et al., 2021; Reynolds & McDonell, 2021), this process is commonly time-intensive and its results are not always desirable in practice. To mitigate such human efforts and achieve better performance in optimizing crafted prompts, random sampling (Zhou et al., 2023), Bayesian optimization (Chen et al., 2023; Lin et al., 2023), and evolutionary algorithms (Guo et al., 2024) have been proposed to generate and select well-performing prompts automatically. However, most of these existing strategies prioritize *global optimization*, dedicating substantial portions of the query budget to explore the entire search space for the global optima and consequently making it query-inefficient in practice. Meanwhile, these strategies typically implement their prompt optimization across various input domains (i.e., natural texts (Guo et al., 2024; Zhou et al., 2023) or hidden embeddings (Chen et al., 2023; Lin et al., 2023)), resulting in diverse performance outcomes in practice. These results consequently inspire us to re-think the questions about the necessity of finding a global optimum and the essence of the input domain for efficient and effective prompt optimization.

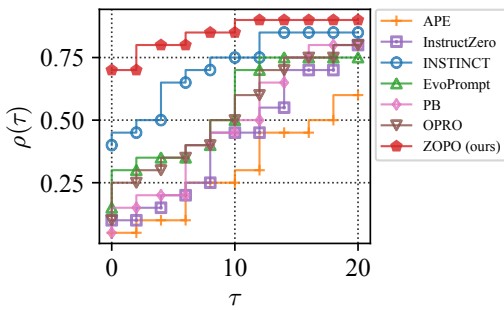

Figure 1: The performance profile for different methods on 20 tasks. A higher $\rho(\tau)$ is better. More details in Sec. 5.

To answer these questions, we provide a thorough empirical study on prompt optimization. Firstly, we visualize the performances of some randomly sampled prompt candidates on various tasks to show that, in contrast to the scarcity of global optima, local optima are commonly prevalent and perform reasonably well, making them more more valuable for query-efficient prompt optimization (**Insight I** in Sec. 3.1). Secondly, we visualize the estimated accuracy distributions for a number of prompt candidates and the corresponding function surfaces using various embeddings as their representation. The results demonstrate that the selection of the input domain, including both the generation and representation of prompt candidates, will influence the identification of high-performing prompts, especially those local optimal ones (**Insight II** in Sec. 3.2). These insights consequently highlight the importance of local optima and input domain for efficient and effective prompt optimization.

Inspired by these insights, we novelly propose the *Localized Zeroth-Order Prompt Optimization* (ZOPO) algorithm for a considerably improved prompt optimization as evidenced by Fig. 1. Motivated by Insight II, we first propose a general domain transformation that utilizes LLMs for prompt generation and existing embedding models for transforming these generated prompts into their corresponding hidden representations, which thereby enjoys not only the remarkable generation ability from any type of LLMs (white/black-box) but also the impressive representation ability from existing embedding models for our prompt optimization (Sec. 4.1). Inspired by Insight I, we then leverage a cutting-edge zeroth-order optimization (ZOO) method enhanced by a derived Gaussian process for efficient gradient estimation (Shu et al., 2023a) to underpin our localized prompt optimization, which goes one step further by incorporating the Neural Tangent Kernel (NTK) (Jacot et al., 2018) to handle the complex and high-dimensional prompt optimization tasks (Sec. 4.2). Lastly, we present an uncertainty-informed local exploration method designed to improve the gradient estimation in our derived NTK-GP, thereby augmenting the practical performance of the ZOPO algorithm (Sec. 4.3).

To summarize, the contributions of our work include:

- To the best of our knowledge, we are the first to conduct a thorough empirical study in prompt optimization to underscore the value of local optima and the essence of input domain for efficient and effective prompt optimization (Sec. 3).
- Drawing on the insights gained from our empirical study, we design the ZOPO algorithm (Sec. 4) which outperforms existing baselines in optimization performance and query efficiency.
- We conduct extensive studies to confirm the efficacy of our algorithmic framework and elucidate the underlying principles or insights of our ZOPO algorithm (Sec. 5).

## 2. Problem Setup

Given an NLP task that is characterized by a data distribution $\mathcal{D}$ and a black-box LLM $f(\cdot)$, e.g., ChatGPT (OpenAI, 2024a), discrete prompt optimization aims to generate a piece of human-readable text, namely the prompt $v$, which will then be applied to the black-box LLM $f(\cdot)$ along with a test input $x$ such that the queried LLM output $f([v; x])$ is able to correctly predict the ground-truth label $y$ for each $(x, y) \sim \mathcal{D}$. This problem is then commonly framed as a black-box maximization problem over the discrete language input domain $\Omega$ (Chen et al., 2023; Lin et al., 2023):

$$\max_{v \in \Omega} F(v) \triangleq \mathbb{E}_{(x,y) \in \mathcal{D}_V} \left[ \mathcal{R} \left( f([v; x]), y \right) \right] \quad (1)$$

where $\mathcal{R} \left( f([v; x]), y \right)$ is applied to measure the alignment between the LLM output $f([v; x])$ and the groundtruth $y$, and $\mathcal{D}_V$ is the validation set sampled from $\mathcal{D}$. Note that the performance of the optimal instruction found on $\mathcal{D}_V$ (i.e., $\arg\max_v F(v)$) will be evaluated on a held-out test set $\mathcal{D}_T$.

## 3. Empirical Study on Prompt Optimization

### 3.1. Local Optima vs. Global Optimum

In prompt optimization, methods like (Chen et al., 2023; Lin et al., 2023) are generally more effective than the others (Zhou et al., 2023; Guo et al., 2024), which is usually contributed to their usage of Bayesian optimization, a popular global optimization strategy, that is able to find the global optimum in low-dimensional problems (Moriconi et al., 2020). However, these methods sometimes perform poorly in certain prompt optimization tasks, e.g., cause_and_effect and informal_to_formal, indicating that they will fail to find the global optimum in these tasks given a limited query budget. This is likely because substantial portions of the budget are applied in these methods to explore the entire search space for the global optimum, which hence leads to the critical question about *the necessity of finding the global optimum in query-efficient prompt optimization*.

To answer this question, we have employed a 3-dimensional scatter plot to visualize the performance (differentiated by

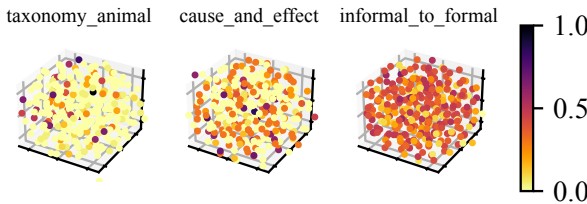

Figure 2: The validation accuracy of 300 randomly sampled prompts with the last token representation on various tasks.

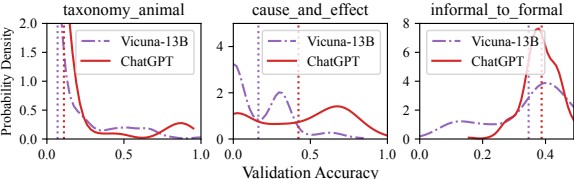

Figure 3: The estimated accuracy distribution of prompts generated by Vicuna-13B or ChatGPT on various instruction induction tasks, where the vertical dotted line indicates the mean performance.

colors) for 300 randomly sampled prompt candidates on various tasks, whose prompt embeddings (i.e., the last token embedding as in (Lin et al., 2023)) are reduced by t-distributed stochastic neighbor embedding (t-SNE) (see more details in our Appx. D.1.1). The results are in Fig. 2 which shows that the global optimum (i.e., the points achieving the highest accuracy) is consistently rare for a range of prompt optimization tasks, making it extremely challenging to achieve this global optimum in practice. In contrast, prompt optimization often features a number of local optima (e.g., the points achieving accuracy higher than 80% in taxonomy_animal of Fig. 2). Importantly, these local optima commonly enjoy relatively good performances, suggesting that local optima shall be more worthwhile to obtain in prompt optimization, especially for the scenarios of limited query budgets, as summarized below.

> **Insight I**
>
> Contrasting with the rarity of global optimum, local optima are usually prevalent and well-performed, which is more worthwhile for query-efficient prompt optimization.

### 3.2. Essence of Input Domain

Besides, existing works (Chen et al., 2023; Guo et al., 2024; Lin et al., 2023) typically implement their prompt optimization across various input domains, leading to a wide range of performances in practice. These results thus inspire us to ask: *How essential is the input domain for finding well-performing prompts, particularly the local optimal ones?* Thoroughly exploring this question is fundamental for the design of a well-performing prompt optimization algorithm.

To answer this, we first visualize the accuracy distributions of 300 prompt candidates that are randomly generated by Vicuna-13B and ChatGPT for various tasks to study the essence of prompt generation in Fig. 3 (more details in Appx. D.1.2). Fig. 3 reveals that the prompt candidates produced by ChatGPT (a black-box model) generally exhibit better performance than those produced by Vicuna-13B (a white-box model), which has been widely applied in (Chen et al., 2023; Lin et al., 2023) for prompt optimization. Importantly, ChatGPT demonstrates a greater likelihood of generating locally optimal prompts (e.g., the ones of accuracy higher

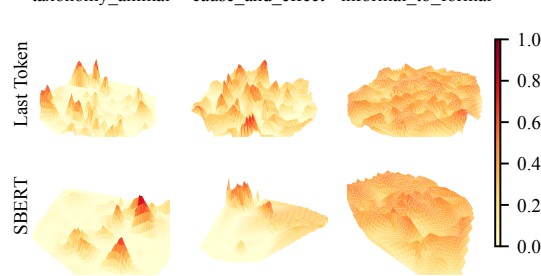

Figure 4: The function surfaces using the last token (Vicuna-13B) or SBERT embedding.

than 0.8 in taxonomy_animal of Fig. 3). These results indicate that the ability to generate well-performing local optima in prompt optimization usually varies for different models. So, the selection of the prompt generation model is crucial for finding well-performing optima.

We then investigate the function surface (i.e., accuracy landscape) using two different embeddings for prompt candidates in Fig. 4 (more details in Appx. D.1.2) where the embeddings are mapped into a 2-dimensional domain using the t-SNE for better visualization. Interestingly, Fig. 4 unveils that different embeddings will convey a varying number of well-performing local optima in practice. Particularly, the last token embedding is usually able to produce a larger number of well-performing local optima than the SBERT (i.e., a popular sentence embedding transformer (Reimers & Gurevych, 2019)) embedding, making it easier to enjoy a good prompt optimization performance on this domain, as validated in Tab. 8. This therefore implies that the choice of the prompt embedding model is also essential for the finding of well-performing optima. In all, we conclude our aforementioned insights as below.

> **Insight II**
>
> The choice of the input domain, covering both the generation and the representation of prompt candidates, affects the identification of well-performing local optima.

**Algorithm 1** The ZOPO Algorithm

1: **Input:** prompt generation model $g(\cdot)$, NLP embedding model $h(\cdot)$, size of prompt candidates $m$, iteration number $T$, set $\mathcal{V} = \emptyset$, set $\mathcal{Z} = \emptyset$
2: **repeat**
3:      $v \leftarrow g([\mathcal{D}_{\text{demo}}])$
4:      $z \leftarrow h(v)$
5:      **if** $v \notin \mathcal{V}$ **then** $\mathcal{V} \leftarrow \mathcal{V} \bigcup \{v\}$, $\mathcal{Z} \leftarrow \mathcal{Z} \bigcup \{z\}$
6: **until** $|\mathcal{V}| = m$
7: **for** $t = 1$ **to** $T$ **do**
8:      **if** $\mathbb{1}_{A_t}(z_t) = 1$ **then** do local exploration in Sec. 4.3
9:      $z_{t+1} = \mathcal{P}_{\mathcal{Z}}(z_t + \eta_t \mu_t(z_t))$
10:      Query $z_{t+1}$ to yield $\widetilde{F}(z_{t+1})$
11: **end for**
12: $z^* \leftarrow \arg\max_{z_{1:T}} \widetilde{F}(z)$
13: **Return** $h^{-1}(z^*)$

## 4. The ZOPO Algorithm

Given the insights established in our Sec. 3, we then propose our *Localized $\underline{Z}$eroth-$\underline{O}$rder $\underline{P}$rompt $\underline{O}$ptimization* (ZOPO) algorithm (Algo. 1) for a better-performing as well as more query-efficient prompt optimization. Specifically, following our Insight II, we first develop a more general transformation for the input domain of prompt optimization (Sec. 4.1), which can enjoy both the remarkable generation ability from any type of LLMs (white/black-box) and the impressive representation ability from many NLP models. Subsequent to this transformation, inspired by our Insight I, we propose to use zeroth-order optimization (ZOO) with a derived NTK Gaussian process inspired from (Shu et al., 2023a) to find well-performing local optima (Sec. 4.2). Lastly, we introduce an uncertainty-informed local exploration technique to refine the gradient estimation in our derived NTK Gaussian process, aiming to enhance the performance of our ZOPO algorithm in practice (Sec. 4.3).

### 4.1. A More General Input Domain Transformation

As introduced in our Sec. 3.2, the choice of input domain (including the generation and representation of candidates) significantly influences the ultimate performance in prompt optimization: Black-box LLMs (e.g., ChatGPT) typically enjoy an advanced generation ability and different embedding models (e.g., SBERT) have varying representative capacity for prompt optimization. This naturally inspires us to develop an improved *domain transformation* that can utilize not only the remarkable generation ability from white/black-box LLMs but also the impressive representation ability from certain NLP models for our prompt optimization. To achieve this, we propose to make use of the prompt $v \in \Omega$ generated from a LLM $g(\cdot)$ and subsequently transform it into a continuous hidden representation $z \in \mathcal{Z} \subset \mathbb{R}^d$ by

other sentence embedding model $h(\cdot)$ for the optimization, i.e., $v = h^{-1}(z)$, where (1) can then be re-framed as

$$\max_{z \in \mathcal{Z}} \widetilde{F}(z) = \mathbb{E}_{(x,y) \in \mathcal{D}} \left[ \mathcal{R}\left(f([h^{-1}(z); x]), y\right) \right] . \quad (2)$$

Of note, our input domain transformation and (2) enjoy a number of major advantages compared with previous works: *(a)* Different from the direct optimization over the discrete and complex language space $v \in \Omega$ in (Guo et al., 2024) where optimization algorithms in the numerical domain can hardly be applied, our transformed input domain leads to a dense numerical space of lower dimension and therefore allows the usage of query-efficient optimization algorithms for (2) (e.g., our Algo. 1). *(b)* Different from the potential many-to-one mapping in the previous works (Chen et al., 2023; Lin et al., 2023), i.e., the same discrete prompt $v$ may be generated by various continuous soft prompts $s$, we develop a one-to-one mapping where one prompt generally has a unique hidden representation $z$, which thus can help eliminate the redundant queries during optimization and ultimately lead to more query-efficient prompt optimization. *(c)* Our domain transformation with an independent generation and representation process is capable of enjoying the remarkable generation ability from any type of LLMs (white/black-box) and the impressive representation ability from many NLP models whereas previous works are highly restricted to the LLMs, thus leading to a wider application.

**Practical Implementations.** Before the start of the optimization on (2), we usually generate numerous prompt *candidates* $\mathcal{V} = \{v\}$ and their corresponding representations $\mathcal{Z} = \{z\}$ (line 2-6 of Algo. 1), where $\mathcal{Z}$ can be produced by an embedding model $h(\cdot)$. We store $(z, v)$ in key-value pairs for constructing the one-to-one inverse mapping $h^{-1}(\cdot)$. Two practical methods are considered here for prompt generation: *(a)* Feeding randomly sampled soft prompts $s \in \mathbb{R}^d$ and a few demonstrations $\mathcal{D}_{\text{demo}}$ into a white-box LLM $g(\cdot)$. *(b)* Sampling the output distribution of a black-box LLM $g(\cdot)$ given a generation template filled with $\mathcal{D}_{\text{demo}}$. Specifically, if we consider the generation method in *(a)*, $z$ can be chosen as the last token embedding from $g(\cdot)$ (Lin et al., 2023) or the soft prompt $s$ (Chen et al., 2023) when generating $v$. Here $h(\cdot)$ then represents a mapping function from $v$ to $z$.

### 4.2. Local Optimization with Derived NTK-GP

As local optima are more prevalent than global optimum and can exhibit compelling performance for prompt optimization tasks (Sec. 3.1), we propose to apply zeroth-order optimization (ZOO), particularly gradient descent using estimated gradients, for a well-performing local prompt optimization on our transformed input domain $\mathcal{Z}$ in Sec. 4.1. Unfortunately, existing ZOO algorithms are typically query-

inefficient as many additional queries are required for gradient estimation in every gradient descent update (Flaxman et al., 2005; Nesterov & Spokoiny, 2017). In light of this, we resort to the most recent ZoRD algorithm (Shu et al., 2023a) where a localized surrogate model will be applied for query-efficient gradient estimations.

According to (Shu et al., 2023a), given a well-specified kernel function $k(\cdot, \cdot)$ such that the function $\widetilde{F}$ is sampled from a Gaussian process $\widetilde{F} \sim \mathcal{GP}(0, k(\cdot, \cdot))$ or alternatively $\min_{G \sim \mathcal{GP}(0,k(\cdot,\cdot))} \max_{z \in \mathcal{Z}} |\widetilde{F}(z) - G(z)| = 0$ and the observed value $r$ of function $\widetilde{F}$ follows the Gaussian noise $\mathcal{N}(0, \sigma^2)$, then conditioned on the history of function queries $\mathcal{D}_t \triangleq \{(z_\tau, r_\tau)\}_{\tau=1}^t$ of size $t$, $\nabla \widetilde{F}$ follows a derived Gaussian Process $\mathcal{GP}(\mu(\cdot), \Sigma(\cdot, \cdot))$, i.e.,

$$\nabla \widetilde{F} \sim \mathcal{GP}\left(\mu_t(\cdot), \Sigma_t^2(\cdot, \cdot)\right), \quad (3)$$

in which the mean function $\mu_t(\cdot)$ and the covariance function $\Sigma_t^2(\cdot, \cdot)$ are defined as

$$\mu_t(z) \triangleq \boldsymbol{k}_t(z)^\top \left(\mathbf{K}_t + \sigma^2 \mathbf{I}\right)^{-1} \boldsymbol{r}_t,$$
$$\Sigma_t^2(z, z') \triangleq k''(z, z') - \boldsymbol{k}_t(z)^\top \left(\mathbf{K}_t + \sigma^2 \mathbf{I}\right)^{-1} \boldsymbol{k}_t(z').$$
$$(4)$$

Here, $\boldsymbol{k}_t(z)^\top \triangleq [\partial_z k(z, z_\tau)]_{\tau=1}^t$ is a $d \times t$-dimensional matrix, $\mathbf{K}_t \triangleq [k(z_\tau, k(z_{\tau'})]_{\tau,\tau'=1}^t$ is a $t \times t$-dimensional matrix, $\boldsymbol{r}_t^\top \triangleq [r_\tau]_{\tau=1}^t$ is a $t$-dimensional column vector, and $k''(z, z') \triangleq \partial_z \partial_{z'} k(z, z')$ is a $d \times d$-dimensional matrix. As a result, $\mu_t(z)$ can be applied to estimate the gradient of the black-box function $\widetilde{F}$ at input $z$.

Of note, the underlying black-box function $\widetilde{F}$ here is highly related to deep neural networks (DNN), more specifically transformers. It naturally inspires us to apply the Neural Tangent Kernel (NTK) (Jacot et al., 2018) theory for a better approach to the aforementioned assumption of a well-specified kernel function $k(\cdot, \cdot)$. This is because it has been widely proven that NTK is capable of well characterizing the predictions of neural networks (Arora et al., 2019; Lee et al., 2019; Shu et al., 2022a;b) and therefore should be a better-specified kernel in the setting of prompt optimization than the simple kernel (i.e., Matérn kernel) applied in ZoRD (Shu et al., 2023a). Specifically, given a neural network $\phi(\theta, z)$ parameterized by $\theta \in \mathbb{R}^p$, we employ the following empirical NTK as the kernel in (3) and (4):

$$k(z, z') = \nabla_\theta \phi(\theta, z)^\top \nabla_\theta \phi(\theta, z)\Big|_{\theta=\theta_0} \quad (5)$$

where $\theta_0$ is the initialized parameter of neural network $\phi$. By incorporating (5) into (4), we realize the derived NTK-GP for the gradient estimation in our prompt optimization.

Based on this derived NTK-GP, we finally apply standard first-order optimization (e.g., stochastic gradient descent) with projected gradients for our local prompt optimization.

Specifically, in every iteration $t$ of our Algo. 1, the next promising prompt candidate will be selected via:

$$v_{t+1} = h^{-1}\left(\mathcal{P}_{\mathcal{Z}}(z_t + \eta_t \mu_t(z_t))\right) \quad (6)$$

where $\mathcal{P}_{\mathcal{Z}}(z) \triangleq \arg\min_{z' \in \mathcal{Z}} \|z - z'\|$ is the projection function that projects the updated $z \in \mathbb{R}^d$ into domain $\mathcal{Z}$ and $\eta_t$ is learning rate.

**Practical Implementations.** Following the localized modeling principle, only the neighbors of $z$ in the query history $\mathcal{D}_t$ are used to calculate the gradient $\mu_t(z)$. As we do not know the exact DNN for the underlying black-box function $\widetilde{F}$, we propose to approximate it using a small DNN, which can work well thanks to the theoretically guaranteed universal approximation ability of DNNs (Kratsios & Papon, 2022; Shen et al., 2022). Our experiments in Sec. 5.4 will further validate the effectiveness of this implementation.

### 4.3. Uncertainty-Informed Local Exploration

Though the derived NTK-GP allows us to estimate the gradient at *any* $z \in \mathcal{Z}$ according to (Shu et al., 2023a), we introduce the following Prop. 4.1 to demonstrate that the error in gradient estimation at a specific input $z \in \mathcal{Z}$ implies considerable variability, which is strongly correlated with the number of historical queries that are *effectively* relevant for the gradient estimation at the specific input $z \in \mathcal{Z}$. This insight, in turn, motivates the creation of our uncertainty-informed local exploration approach, as opposed to the adoption of the virtual update mechanism described in (Shu et al., 2023a) for our prompt optimization strategy.

**Proposition 4.1.** *Assume $k(z, z') \leq \alpha$ and $\|k''(z, z)\| \leq \kappa$ for any $z, z' \in \mathcal{Z}$. Let $\delta \in (0, 1)$ and $N_{z,\beta} \triangleq \{z' \in \{z_\tau\}_{\tau=1}^t \mid \|\partial_z k(z', z)\|^2 \geq \beta\}$ for given input $z \in \mathcal{Z}$, the following holds with a probability of at least $1 - \delta$,*

$$\|\mu_t(z) - \nabla F(z)\|^2 \leq \omega \left\|\Sigma_t^2(z)\right\| \leq \omega\kappa - \frac{\omega\beta/d}{\alpha + \sigma^2/|N_{z,\beta}|}$$

*where $\omega = d + 2(\sqrt{d} + 1)\ln(1/\delta)$ and $\Sigma_t^2(z) \triangleq \Sigma_t^2(z, z)$.*

The proof is given in Appx. A. Here, $N_{z,\beta}$ denotes a set of historical input queries that are effectively relevant for the gradient estimation at $z$ where $\beta$ can be regarded as a measure of effective relevance. Prop. 4.1 shows that the gradient estimation error of (3) at a specific input $z \in \mathcal{Z}$ is bounded by the norm of covariance matrix $\Sigma_t^2(z)$, which is related to the query set $N_{z,\beta}$ of effective relevance. Specifically, the gradient estimation error at different $z$ varies if the effective relevance $\beta$ and the number of relevant queries $|N_{z,\beta}|$ varies with $z$. When $\beta$ or $|N_{z,\beta}|$ becomes small during ZOO, the gradient estimation error is likely increased, which will lead to poor performance in practice. This likely will happen in prompt optimization especially considering the sparsity of

prompt candidates w.r.t. the continuous domain $\mathbb{R}^d$. That is, both the effective relevance $\beta$ and the number of relevant queries $|N_{z,\beta}|$ can be small due to this sparsity. As a consequence, additional input queries should be conducted to increase both $\beta$ and $|N_{z,\beta}|$ for better optimization.

To this end, we propose an uncertainty-informed local exploration method that utilizes additional input queries from local searches to reduce predictive uncertainty and hence the gradient estimation error in derived NTK-GP according to Prop. 4.1. Specifically, we propose the local exploration condition informed by the local trajectory:

$$\mathbb{1}_{A_t}(z_t) = \left\{ \begin{array}{ll} 1 & z_t \in A_t \\ 0 & z_t \notin A_t \end{array} \right.$$

where $A_t = \{z_t | \sigma(z_{t-i}) \geq \lambda, \ \forall i \in [0, \xi]\}$ is the condition that incorporates uncertainties and $\lambda, \xi$ are the thresholds. If this condition is met (i.e., $\mathbb{1}_{A_t}(z_t) = 1$), we will query the neighbors of $z_t$ in the local region to update our derived NTK-GP, thus improving its gradient estimation.

**Practical Implementations.** If we define the set of the $n$ nearest neighbors of $z_t$ as $\mathcal{N}_t \subseteq \mathcal{Z}$ s.t. $|\mathcal{N}_t| = n$ and $\forall a \in \mathcal{Z} \setminus \mathcal{N}_t, \|a - z_t\| \geq \max_{b \in \mathcal{N}_t} \|b - z_t\|$, we propose to query each $z \in \mathcal{N}_t$ in the local region, whenever $\mathbb{1}_{A_t}(z_t) = 1$.

## 5. Experiments

In this section, we perform prompt optimization for Chat-GPT (i.e., $f(\cdot)$) and evaluate the performance of ZOPO against several strong baselines, including APE (Zhou et al., 2023), InstructZero (Chen et al., 2023), INSTINCT (Lin et al., 2023), EvoPrompt (Guo et al., 2024), PromptBreeder (PB) (Fernando et al., 2023), and OPRO (Yang et al., 2024), on 30 instruction induction tasks (Honovich et al., 2023), 3 arithmetic reasoning tasks (Cobbe et al., 2021; Ling et al., 2017; Patel et al., 2021), and the GLUE benchmark (Wang et al., 2019). The task-specific prompt is optimized for each task independently. We use the performance profile (Dolan & Moré, 2002), defined in Appx. C.1, as the overall evaluation metric that measures the frequency (i.e., $\rho(\tau)$) of a method within some distance (i.e., $\tau$) from the highest accuracy achieved by any method. We defer more experimental details to Appx. C.

### 5.1. Instruction Induction

Instruction induction tasks are commonly used to investigate the prompt optimization performance by assessing LLM's zero-shot ICL ability in previous works (Chen et al., 2023; Lin et al., 2023; Zhou et al., 2023). Although our ZOPO is a general prompt optimization method given any prompt generation strategy, here we follow the same setting of prompt generation from INSTINCT and InstructZero, only for **fair comparison**. We also adopt the last token embedding from

Vicuna-13B as the prompt embedding (same as INSTINCT). Here Vicuna-13B is used to generate task-specific prompts by feeding random soft prompts. More experimental details are deferred to Appx. C.3.

**Superior performance of ZOPO.** For better distinguishability, we follow the experimental setting from Lin et al. (2023) to display the results on 20 challenging tasks reported in Tab. 1, where ZOPO significantly outperforms all baseline methods. Particularly, our ZOPO performs the best in 14 out of the 20 tasks presented, while achieving the best performance profile across different $\tau$ (see Fig. 1) compared with all baseline methods. For more results on all 30 tasks, refer to Tab. 3 in Appx. D.2, where the ZOPO consistently outperforms existing methods.

**ZOPO has better query efficiency.** To justify that our local optimization method is more *query-efficient*, we compare ZOPO against baselines at different query budget scales. The results shown in Fig. 5 and Fig. 10 in Appx. D.2 illustrate that ZOPO generally achieves better performance with the same number of queries compared with other baseline methods and yields superior performance upon convergence. We notice that ZOPO achieves lower validation accuracy yet higher test accuracy on the `taxonomy_animal` task than INSTINCT, which suggests ZOPO likely has better generalization ability.

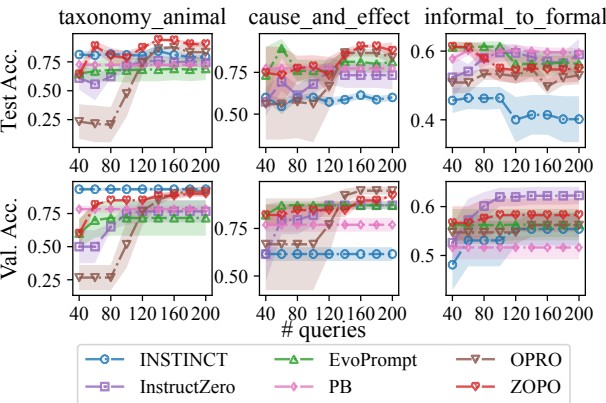

Figure 5: Comparison of the query efficiency between ZOPO and baselines. The first and second rows show the test and validation accuracies.

**Connecting ChatGPT with ZOPO.** With our proposed domain transformation, we empirically demonstrate that ZOPO is capable of performing numerical optimization on ChatGPT-generated prompts. Specifically, we use the same generation method as in APE (Zhou et al., 2023) to generate task-specific prompts (i.e., $\mathcal{V}$) from ChatGPT, and use a popular embedding model SBERT to provide the corresponding sentence embeddings (i.e., $\mathcal{Z}$) for $\mathcal{V}$. Then we apply ZOPO to perform optimization over the given $\mathcal{V}$ and

Table 1: Average test accuracy with standard error (3 runs) for different methods on 20 instruction induction tasks. We **bold** the highest accuracy when comparing ZOPO with baselines, and use green cell to highlight the highest accuracy when comparing ZOPO$_{\text{GPT}}$ with baselines.

| Tasks | APE | InstructZero | INSTINCT | EvoPrompt | PB | OPRO | ZOPO | ZOPO$_{\text{GPT}}$ |
|---|---|---|---|---|---|---|---|---|
| antonyms | $63.7_{\pm14.2}$ | $82.7_{\pm0.7}$ | $84.7_{\pm0.3}$ | $84.0_{\pm0.0}$ | $78.0_{\pm3.6}$ | $79.0_{\pm2.2}$ | $\mathbf{85.2}_{\pm3.2}$ | $84.0_{\pm1.4}$ |
| auto_categorization | $25.0_{\pm0.9}$ | $25.7_{\pm1.2}$ | $25.0_{\pm3.3}$ | $31.0_{\pm1.0}$ | $24.0_{\pm3.7}$ | $24.0_{\pm3.6}$ | $\mathbf{32.7}_{\pm1.9}$ | $27.0_{\pm5.0}$ |
| auto_debugging | $29.2_{\pm3.4}$ | $37.5_{\pm0.0}$ | $29.2_{\pm3.4}$ | $33.0_{\pm7.2}$ | $25.0_{\pm0.0}$ | $37.5_{\pm0.0}$ | $\mathbf{41.7}_{\pm15.6}$ | $29.2_{\pm5.9}$ |
| cause_and_effect | $57.3_{\pm8.9}$ | $81.3_{\pm1.1}$ | $58.7_{\pm8.7}$ | $84.0_{\pm13.9}$ | $82.7_{\pm10.0}$ | $82.7_{\pm10.0}$ | $\mathbf{94.7}_{\pm3.7}$ | $80.0_{\pm14.2}$ |
| common_concept | $6.9_{\pm2.1}$ | $8.6_{\pm4.0}$ | $21.3_{\pm0.2}$ | $11.1_{\pm6.9}$ | $10.9_{\pm3.4}$ | $8.6_{\pm5.7}$ | $\mathbf{23.5}_{\pm3.4}$ | $2.8_{\pm0.6}$ |
| diff | $67.3_{\pm26.7}$ | $69.3_{\pm22.2}$ | $\mathbf{100.0}_{\pm0.0}$ | $27.3_{\pm42.2}$ | $71.3_{\pm27.6}$ | $\mathbf{100.0}_{\pm0.0}$ | $\mathbf{100.0}_{\pm0.0}$ | $100.0_{\pm0.0}$ |
| informal_to_formal | $57.4_{\pm0.3}$ | $53.1_{\pm0.2}$ | $55.3_{\pm0.0}$ | $51.6_{\pm0.9}$ | $54.2_{\pm4.5}$ | $48.0_{\pm0.8}$ | $\mathbf{61.3}_{\pm2.7}$ | $61.9_{\pm2.9}$ |
| letters_list | $100.0_{\pm0.0}$ | $59.0_{\pm16.7}$ | $100.0_{\pm0.0}$ | $100.0_{\pm0.0}$ | $99.3_{\pm0.9}$ | $99.7_{\pm0.5}$ | $\mathbf{100.0}_{\pm0.0}$ | $100.0_{\pm0.0}$ |
| negation | $75.3_{\pm1.1}$ | $77.7_{\pm1.4}$ | $81.7_{\pm0.3}$ | $86.0_{\pm0.0}$ | $70.7_{\pm4.0}$ | $73.3_{\pm6.6}$ | $\mathbf{86.3}_{\pm0.5}$ | $77.7_{\pm2.6}$ |
| object_counting | $36.3_{\pm1.9}$ | $36.0_{\pm9.3}$ | $34.0_{\pm7.0}$ | $\mathbf{55.0}_{\pm5.3}$ | $29.3_{\pm1.2}$ | $36.0_{\pm5.7}$ | $52.3_{\pm6.6}$ | $40.3_{\pm0.5}$ |
| odd_one_out | $63.3_{\pm1.4}$ | $61.3_{\pm8.7}$ | $\mathbf{70.0}_{\pm1.6}$ | $10.0_{\pm0.0}$ | $66.7_{\pm0.9}$ | $47.3_{\pm22.2}$ | $32.0_{\pm11.3}$ | $68.7_{\pm2.5}$ |
| orthography_starts_with | $45.7_{\pm14.8}$ | $50.7_{\pm8.7}$ | $\mathbf{66.7}_{\pm2.7}$ | $15.0_{\pm3.4}$ | $59.8_{\pm5.7}$ | $33.5_{\pm14.6}$ | $56.5_{\pm12.6}$ | $71.0_{\pm0.0}$ |
| rhymes | $15.7_{\pm6.4}$ | $100.0_{\pm0.0}$ | $100.0_{\pm0.0}$ | $59.7_{\pm3.1}$ | $45.0_{\pm10.7}$ | $23.0_{\pm14.7}$ | $\mathbf{100.0}_{\pm0.0}$ | $61.0_{\pm2.8}$ |
| second_word_letter | $\mathbf{74.7}_{\pm20.3}$ | $43.3_{\pm18.7}$ | $10.0_{\pm4.1}$ | $24.7_{\pm0.6}$ | $88.7_{\pm10.4}$ | $86.7_{\pm18.9}$ | $25.7_{\pm4.7}$ | $96.7_{\pm2.4}$ |
| sentence_similarity | $0.0_{\pm0.0}$ | $0.0_{\pm0.0}$ | $\mathbf{14.0}_{\pm0.5}$ | $2.0_{\pm1.0}$ | $0.0_{\pm0.0}$ | $2.7_{\pm3.8}$ | $7.6_{\pm9.3}$ | $37.3_{\pm0.9}$ |
| sum | $67.3_{\pm26.7}$ | $100.0_{\pm0.0}$ | $100.0_{\pm0.0}$ | $100.0_{\pm0.0}$ | $98.3_{\pm1.7}$ | $\mathbf{100.0}_{\pm0.0}$ | $\mathbf{100.0}_{\pm0.0}$ | $100.0_{\pm0.0}$ |
| synonyms | $36.0_{\pm7.6}$ | $27.7_{\pm9.3}$ | $30.7_{\pm4.9}$ | $40.3_{\pm4.0}$ | $36.3_{\pm3.3}$ | $40.0_{\pm4.3}$ | $\mathbf{43.3}_{\pm0.9}$ | $44.7_{\pm4.1}$ |
| taxonomy_animal | $34.7_{\pm23.4}$ | $71.7_{\pm8.4}$ | $85.7_{\pm6.0}$ | $83.0_{\pm4.6}$ | $29.7_{\pm38.5}$ | $30.0_{\pm41.0}$ | $\mathbf{90.0}_{\pm7.1}$ | $92.3_{\pm0.5}$ |
| word_sorting | $33.0_{\pm3.7}$ | $31.0_{\pm11.4}$ | $51.3_{\pm0.3}$ | $48.0_{\pm21.3}$ | $45.7_{\pm1.7}$ | $50.3_{\pm5.8}$ | $\mathbf{60.0}_{\pm4.2}$ | $60.3_{\pm3.1}$ |
| word_unscrambling | $44.0_{\pm13.9}$ | $55.0_{\pm1.7}$ | $63.3_{\pm0.7}$ | $51.3_{\pm4.5}$ | $51.0_{\pm6.2}$ | $61.3_{\pm2.1}$ | $59.3_{\pm2.8}$ | $58.3_{\pm1.9}$ |

$\mathcal{Z}$, which we name ZOPO$_{\text{GPT}}$. The result of ZOPO$_{\text{GPT}}$ compared against other baselines is shown in Tab. 1, with the corresponding performance profile shown in Fig. 9 in App. D.2. Fig. 9 demonstrates that ZOPO$_{\text{GPT}}$ significantly outperforms other baselines, achieving the best performance in 10 out of the 20 tasks as shown in Tab. 1. Specifically, ZOPO$_{\text{GPT}}$ achieves significantly higher accuracy on some challenging tasks such as `second_word_letter` and `sentence_similarity`, which we attribute to the high-quality of prompt candidates generated by ChatGPT. This is also consistent with our discussion on the input domain in Sec. 3.2. Here we could not draw a direct comparison between ZOPO and ZOPO$_{\text{GPT}}$, as the Vicuna last token embedding is specifically associated with the prompt generation process in ZOPO and cannot be applied to ZOPO$_{\text{GPT}}$. However, using either ZOPO or ZOPO$_{\text{GPT}}$ is sufficient to outperform baseline methods, which also provides the flexibility of prompt optimization in practice. Future research may consider employing better embeddings to further improve the performance of ZOPO$_{\text{GPT}}$.

### 5.2. Improving Chain-of-Thought Prompt

The hand-crafted prompt "Let's think step by step" (Kojima et al., 2022) (denoted as hand-craft) has been shown effective in improving LLMs' zero-shot multi-step reasoning performance. We show that ZOPO can find a better chain-of-thought prompt across different arithmetic reasoning tasks, as evidenced in Tab. 2 in Appx. C.4. Particularly, ZOPO produces a better prompt "Let's find the solution by using the given information." on GSM8K (Cobbe et al.,

2021) compared to other baselines, improving the performance from 71.8 (hand-craft) to 75.4. Refer to Appx. C.4 for more experimental details.

### 5.3. Ablation Study

**Verifying the essence of input domain.** To fairly validate the importance of input domain on prompt generation, we compare the optimization performances with different prompts generated by Vicuna-13B and ChatGPT respectively, using the same embedding model SBERT (i.e., $h(\cdot)$). The result is shown in Table. 7 in Appx. D.6, with the performance profile in Fig. 11 suggesting that applying ZOPO on ChatGPT-generated prompts is better. We ascribe its better performance to ChatGPT's remarkable prompt generation ability. This confirms the importance of the input domain on prompt generation in our Insight II.

Besides, different embeddings (i.e., $\mathcal{Z}$) of the same prompt candidates can potentially affect the function landscape as shown in Fig. 4. Thus, we need to study the performance of ZOPO using different embedding representations given the same set of prompts. We consider four different embeddings here: the last token embedding from Vicuna-13B, the OpenAI embedding provided through an API (OpenAI, 2024b), the SBERT embedding, and a randomly projected embedding baseline. We observe from Tab. 8 in Appx. D.6 that, although last token embedding is generally better, there are certain tasks that OpenAI and SBERT embeddings perform equally well or better. Besides, random embedding shows a distinct lesser performance. This again highlights the im-

portance of using more structured embeddings for prompt optimization and indicates the optimal choice of embedding can be *task-dependent*. We discuss how we might find better embeddings and further show the generality of ZOPO by experimenting with more embedding models in Appx. D.6.

**Study of NTK-GP and uncertainty-informed local exploration.** We conducted additional experiments to validate the NTK-GP (Sec. 4.2) and uncertainty-informed local exploration (Sec. 4.3) components of ZOPO. We evaluated the impact of these components by testing two variants of the ZOPO algorithm: (a) replacing the NTK component with Matérn kernel (as in ZoRD), and (b) removing the uncertainty-informed local exploration. Comparisons of these variants against the original ZOPO on instruction induction tasks (see Tab. 11 in Appx. D.7) highlight the significant contributions of these components to ZOPO's overall effectiveness.

**Additional results.** The results on the GLUE benchmark in Appx. D.3 consistently validate the superior performance of ZOPO. We also demonstrate that ZOPO can handle prompt optimization in the *few-shot* ICL setting in Appx. D.4. We conduct further experiments to show ZOPO generalize to *different combinations* of prompt generation models and black-box LLMs in Appx. D.5. We also perform an ablation study to examine the impact of a larger size of the generated prompt candidates (i.e., $|\mathcal{V}|$) on ZOPO and ZOPO$_{\text{GPT}}$ in Appx. D.8, which suggests a relatively small set of strong prompt candidates (e.g., $|\mathcal{V}| = 500$) is sufficient (compared with size 1000 or 2000). Additionally, we provide more demonstrations of our empirical findings in Sec. 3 on other tasks in Appx. D.1.

### 5.4. Ablation Study

**Verifying the essence of input domain.** To fairly validate the importance of input domain on prompt generation, we compare the optimization performances with different prompts generated by Vicuna-13B and ChatGPT respectively, using the same embedding model SBERT (i.e., $h(\cdot)$). The result is shown in Table. 7 in Appx. D.6, with the performance profile in Fig. 11 suggesting that applying ZOPO on ChatGPT-generated prompts is better. We ascribe its better performance to ChatGPT's remarkable prompt generation ability. This confirms the importance of the input domain on prompt generation in our Insight II.

Besides, different embeddings (i.e., $\mathcal{Z}$) of the same prompt candidates can potentially affect the function landscape as shown in Fig. 4. Thus, we need to study the performance of ZOPO using different embedding representations given the same set of prompts. We consider four different embeddings here: the last token embedding from Vicuna-13B, the OpenAI embedding (OpenAI, 2024b), the SBERT embedding, and a randomly projected embedding baseline. We

observe from Tab. 8 in Appx. D.6 that, although last token embedding is generally better, there are certain tasks that OpenAI and SBERT embeddings perform equally well or better. Besides, random embedding shows a distinct lesser performance. This again highlights the importance of using more structured embeddings for prompt optimization and indicates the optimal choice of embedding can be *task-dependent*. We discuss how we might find the best embedding model and further show the generality of ZOPO by experimenting with more embedding models in Appx. D.6.

**Study of NTK-GP and uncertainty-informed local exploration.** We conducted additional experiments to validate the NTK-GP (Sec. 4.2) and uncertainty-informed local exploration (Sec. 4.3) components of ZOPO. We evaluated the impact of these components by testing two variants of the ZOPO algorithm: (a) replacing the NTK component with Matérn kernel (as in ZoRD), and (b) removing the uncertainty-informed local exploration. Comparisons of these variants against the original ZOPO on instruction induction tasks (see Tab. 11 in Appx. D.7) highlight the significant contributions of these components to ZOPO's overall effectiveness.

**Additional results.** The results on the GLUE benchmark in Appx. D.3 consistently validate the superior performance of ZOPO. We also demonstrate that ZOPO can handle prompt optimization in the *few-shot* setting in Appx. D.4. We conduct further experiments to show ZOPO generalize to *different combinations* of prompt generation models and black-box LLMs in Appx. D.5. We also perform an ablation study to examine the impact of a larger size of the generated prompt candidates (i.e., $|\mathcal{V}|$) in Appx. D.8, which suggests a relatively small set of strong prompt candidates (e.g., $|\mathcal{V}| = 500$) is sufficient (compared with size 1000 or 2000). Additionally, we provide more demonstrations of our empirical findings in Sec. 3 on other tasks in Appx. D.1.

## 6. Conclusion

In this work, we first provide a thorough empirical study to understand the characteristics of the target function, and then propose our ZOPO algorithm for prompt optimization. ZOPO embraces a ZOO approach in pursuit of finding local optima efficiently. Extensive experiments on instruction induction tasks, reasoning tasks, and GLUE benchmark demonstrate the efficacy of ZOPO, and ablation studies also validate the design and generality of ZOPO. Besides, we propose a domain transformation that connects powerful LLMs with remarkable embedding models, which provides the flexibility of choices of input domains in prompt optimization. A limitation of this paper is the lack of principle to select LLMs and embedding models in our input domain transformation for better-performing prompt optimization, which we aim to explore in future work.

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

# A. Proofs

## A.1. Proof of Prop. 4.1

We follow the ideas in (Shu et al., 2023a;b) to prove our Prop. 4.1. To begin with, we first introduce the following lemmas adapted from (Shu et al., 2023a):

**Lemma A.1** (Thm. 1 in (Shu et al., 2023a)). *Let* $\delta \in (0, 1)$ *and* $\omega \triangleq d + 2(\sqrt{d} + 1) \ln(1/\delta)$. *For any* $z \in \mathcal{Z}$ *and any* $t \geq 1$, *the following holds with probability of at least* $1 - \delta$,

$$\left\| \nabla \widetilde{F}(z) - \mu_t(z) \right\|^2 \leq \omega \left\| \Sigma_t^2(z) \right\| .$$

**Lemma A.2** (Lemma B.4 in (Shu et al., 2023a)). *For any* $z \in \mathcal{Z}$ *and any* $t \geq 1$, *the following holds*

$$\left\| \Sigma_t^2(z) \right\| \leq \left\| \Sigma_{t-1}^2(\boldsymbol{x}) \right\| .$$

*Proof of Prop. 4.1.* Recall that the covariance function (refer to (4)) of our derived NTK-GP conditioned on the history of function queries $\mathcal{D}_t \triangleq \{(z_\tau, r_\tau)\}_{\tau=1}^t$ of size $t$ will be

$$\Sigma_t^2(z) = k''(z, z) - \boldsymbol{k}_t(z)^\top \left( \mathbf{K}_t + \sigma^2 \mathbf{I} \right)^{-1} \boldsymbol{k}_t(z) . \tag{7}$$

For any $c \in \mathbb{R}$ and $z \in \mathcal{Z}$, define $N_{z,\beta} \triangleq \{z' \in \{z_\tau\}_{\tau=1}^t \mid \|\partial_z k(z, z')\|^2 \geq \beta\}$ with $|N_{z,\beta}| = N$, the following then holds on the set $N_{z,\beta}$:

$$
\begin{aligned}
\left\| \boldsymbol{k}_N(z)^\top \boldsymbol{k}_N(z) \right\| &\overset{(a)}{\geq} \frac{1}{d} \operatorname{tr} \left( \boldsymbol{k}_N(z)^\top \boldsymbol{k}_N(z) \right) \\
&\overset{(b)}{=} \frac{1}{d} \operatorname{tr} \left( \boldsymbol{k}_N(z) \boldsymbol{k}_N(z)^\top \right) \\
&\overset{(c)}{=} \frac{1}{d} \sum_{n=1}^N \left\| \partial_z k(z, z') \right\|^2 \\
&\overset{(d)}{\geq} \frac{N\beta}{d}
\end{aligned}
\tag{8}
$$

where $(a)$ comes from the fact the maximum eigenvalue of a matrix is always larger or equal to its averaged eigenvalues, $(b)$ is based on $\operatorname{tr}(AB) = \operatorname{tr}(BA)$, $(c)$ is from the definition of $\boldsymbol{k}_N(z)$, and $(d)$ results from the definition of $N_{z,\beta}$.

Meanwhile,

$$
\begin{aligned}
\Sigma_t^2(z) &\overset{(a)}{\preccurlyeq} k''(z, z) - \boldsymbol{k}_N(z)^\top \left( \mathbf{K}_N + \sigma^2 \mathbf{I} \right)^{-1} \boldsymbol{k}_N(z) \\
&\overset{(b)}{\preccurlyeq} \kappa \mathbf{I} - \left( \lambda_{\max} (\mathbf{K}_N) + \sigma^2 \right)^{-1} \boldsymbol{k}_N(z)^\top \boldsymbol{k}_N(z) \\
&\overset{(c)}{\preccurlyeq} \kappa \mathbf{I} - \frac{\boldsymbol{k}_N(z)^\top \boldsymbol{k}_N(z)}{N\alpha + \sigma^2} \\
&\overset{(d)}{\preccurlyeq} \left( \kappa - \frac{N\beta/d}{N\alpha + \sigma^2} \right) \mathbf{I}
\end{aligned}
\tag{9}
$$

where $(a)$ comes from Lemma A.2, $(b)$ is based on the assumption of $\|k''(z, z)\| \leq \kappa$ and the definition of maximum eigenvalue. In addition, $(c)$ comes from $\lambda_{\max}(\mathbf{K}_N) \leq N \max_{z,z' \in N_{z,\beta}} k(z, z')$ (i.e., the Gershgorin theorem) and the assumption that $k(z, z') \leq \alpha$ for any $z, z' \in \mathcal{Z}$, and $(d)$ is based on the results in (8).

Finally, by introducing the results above into Lemma A.1, we conclude the proof.

# B. Broader Impacts

As LLMs have recently received great popularity in human society and their various applications have significantly affected many aspects of society, it is important to make sure the technology related to LLMs is helpful and harmless. Our work focuses on improving the performance of black-box LLMs by automatically optimizing the prompts, which can significantly save human efforts in prompt engineering. However, such work can be potentially used for malicious purposes. When an adversarial user defines a harmful objective function, our work could be exploited to output harmful prompts that lead to negative societal impacts. Therefore, we urge the black-box LLM API providers to impose a security check for the prompt that prevents users from querying for malicious purposes.

# C. Details of Experimental Settings

## C.1. Evaluation Metrics

Following previous works (Zhou et al., 2023; Lin et al., 2023), we use the F1 score for tasks including *common_concept* and *informal_to_formal*; we use the exact set matching for *orthography_starts_with* and *taxonomy_animal*; we use the set containing for *synonyms*; we use the exact matching metric for the rest of instruction induction tasks; and we use the accuracy metric for the arithmetic reasoning datasets.

As the number of datasets is tremendous, we use the performance profile (Dolan & Moré, 2002) as the evaluation metric that measures the frequency (i.e., $\rho(\tau)$) of a method within some distance (i.e., $\tau$) from the optimality achieved by any method, defined below

$$\rho_m(\tau) = \frac{1}{|\Pi|} \left| \{ \pi \in \Pi : r_\pi^* - r_{\pi,m} \leq \tau \} \right| \tag{10}$$

where $\Pi$ is the set of all tasks, $r_{\pi,m}$ is the accuracy of method $m$ on task $\pi$, and $r_\pi^* = \max\{r_{\pi,m} : \forall m \in \mathcal{M}\}$ is the best performance achieved by any method in $\mathcal{M}$ on task $\pi$. Specifically, $\rho(0)$ represents the number of tasks where a method achieves the best performance. Accordingly, we use both $\rho(0)$ and $\rho(5)$ as the evaluation indicators in our tables to report the results.

## C.2. Hyperparameters

For all experiments using ZOPO in this work, we set the learning rate to 0.01, the uncertainty thresholds $\lambda, \xi$ to 0.1 and 5 respectively, and the number $n$ of nearest neighbors to query in local exploration (Section 4.3) to 10. A neural network with 2 fully connected layers of size 32 and ReLU activation functions is used in NTK-GP as the kernel. We use 20 nearest neighbors to fit the NTK-GP.

## C.3. Instruction induction

In this subsection, we describe the experimental details of the instruction induction tasks.

### C.3.1. EXPERIMENTAL SPECIFICATIONS

The same data partition and evaluation process as in previous works (Zhou et al., 2023; Chen et al., 2023; Lin et al., 2023) is adopted in this work, where, for each task, we optimize the generated prompt on a training set $\mathcal{D}$, and report the best-performing prompt's inference accuracy on a held-out *test set* $\mathcal{D}_T$. Specifically, 5 examples are sampled from the training set as the demonstrations (i.e., $\mathcal{D}_{\text{demo}}$) for instruction induction, and another sampled 20 examples from the training set are used as the *validation set* $\mathcal{D}_V$ to evaluate the objective function value as in Equation (1). The total query budget for each instruction induction task is fixed at 165 for all methods.

### C.3.2. IMPLEMENTATION DETAILS

To comprehensively compare with the baseline methods, we use GPT-3.5-turbo-0301 (supported by OpenAI API) as the black-box model for prompt evaluation and Vicuna-13B-v1.1 as the white-box LLM (i.e., $g(\cdot)$) to generate the task-specific prompts by feeding $g(\cdot)$ with randomly sampled soft prompts and $\mathcal{D}_{\text{demo}}$, which is the same as InstructZero and INSTINCT. In the experiments, we only generate 500 prompt candidates for ZOPO (i.e., $|\mathcal{V}| = 500$). Similarly, we also use 40 out of the 165 queries for random initialization of our optimization method, which could serve as the only global exploration of the function landscape at the beginning of local optimization.

To tackle the high dimensionality of soft prompt (i.e., 5120 for one token embedding as in Vicuna-13B) in optimization, InstructZero and INSTINCT use random projection to project the soft prompt into a much smaller intrinsic dimension (e.g., 100). This intrinsic dimension may empirically affect the quality of generated prompts, as shown in Lin et al. (2023). Therefore, tuning the intrinsic dimension and the soft token length could lead to better performance. Previous methods (i.e., InstructZero and INSTINCT) perform a grid search over the intrinsic dimension in {50, 100, 200} and the soft token length {3, 5, 10} on the validation set and report the accuracy on a held-out test set using the best prompt found using the validation set. We also adopt this technique in ZOPO here for fair comparison. The soft prompt will be concatenated with the tokenized embedding of the prompt generation template to generate task-specific prompt from Vicuna-13B. The prompt generation template and the prompt evaluation template are shown below in the bounding boxes.

---

**Prompt Generation Template (Soft Prompt)**

**Input**: ⟨**INPUT**⟩
**Output**: ⟨**OUTPUT**⟩
**Input**: ⟨**INPUT**⟩
**Output**: ⟨**OUTPUT**⟩
**Input**: ⟨**INPUT**⟩
**Output**: ⟨**OUTPUT**⟩
**Input**: ⟨**INPUT**⟩
**Output**: ⟨**OUTPUT**⟩
**Input**: ⟨**INPUT**⟩
**Output**: ⟨**OUTPUT**⟩
**The prompt was to?**

---

**Evaluation Template**

**prompt**: ⟨**prompt** $(i.e., v)$⟩
**Input**: ⟨**TEST INPUT**⟩
**Output**:

---

We directly use the reported results of APE, IntructZero, and INSTINCT from Lin et al. (2023) for comparison, and we report the results of EvoPrompt with our re-implementation. For a fair comparison, we also use Vicuna-13B for generating the initial prompt population (of size 20) for EvoPrompt, and we use GPT-3.5 turbo to perform the genetic algorithm in EvoPrompt and generate its new prompts. Using GPT-3.5 turbo to generate new prompts will help improve EvoPrompt's performance, as compared with using the relatively smaller model Vicuna-13B.

### C.3.3. EXPERIMENTAL DETAILS ON QUERY EFFICIENCY

To facilitate a more comprehensive comparison of different prompt optimization methods at different query budget scales, we set the maximum query budget to 200, and report the test accuracy of the best prompt found on the validation set with each incremental query budget, as shown in Fig. 5 in the main text. We report the mean accuracy and standard error, using 3 runs with different random seeds. For InstructZero, INSTINCT, and ZOPO, we directly fix the intrinsic dimension for generating the soft prompt as 10 and the number of soft tokens as 5, without using the validation set to perform a grid search over the intrinsic dimension and the number of soft tokens.

> **ChatGPT Prompt Generation Template**
>
> **I gave a friend an prompt. Based on the prompt they produced the following input-output pairs:**
>
> **Input**: ⟨**INPUT**⟩
> **Output**: ⟨**OUTPUT**⟩
> **Input**: ⟨**INPUT**⟩
> **Output**: ⟨**OUTPUT**⟩
> **Input**: ⟨**INPUT**⟩
> **Output**: ⟨**OUTPUT**⟩
> **Input**: ⟨**INPUT**⟩
> **Output**: ⟨**OUTPUT**⟩
> **Input**: ⟨**INPUT**⟩
> **Output**: ⟨**OUTPUT**⟩
>
> **The prompt was to**

### C.3.4. EXPERIMENTAL DETAILS ON ZOPO$_{\text{GPT}}$

For our experiment on ZOPO$_{\text{GPT}}$ in the main text, we apply ZOPO on ChatGPT (i.e., GPT-3.5 turbo) generated prompts. We follow the generation template from APE (Zhou et al., 2023), as shown above, to generate task-specific prompts from ChatGPT. To generate various prompts using the APE method, we need to sample different sets of demonstrations (i.e., $\mathcal{D}_{\text{demo}}$) from the training set, and, for each $\mathcal{D}_{\text{demo}}$, we also need to randomly sample from the ChatGPT's response by setting a high temperature (e.g., 0.95). To maintain the same size of prompt candidates as in the previous experimental setting of ZOPO, we also generate 500 prompt candidates for each instruction induction task. To harness the representation power of existing embedding models, we adopt the sentence transformer model (Reimers & Gurevych, 2019) "all-mpnet-base-v2" from HuggingFace to generate the high-dimensional sentence embedding for each generated prompt from ChatGPT.

### C.4. Improving Chain-of-Thought Prompt

To improve the zero-shot chain-of-thought prompt performance on arithmetic reasoning tasks, we make use of the LLM's induction ability and enable LLMs to generate different chain-of-thought prompt candidates by providing some example chain-of-thought prompts. We consider the evaluation of our method on three arithmetic reasoning datasets (i.e., GSM8K(Cobbe et al., 2021), AQUARAT(Ling et al., 2017), SVAMP(Patel et al., 2021)). Similar as APE (Zhou et al., 2023), we use all data from the test set for GSM8K and AQUARAT, and we sample 400 data points from AQUARAT's test set to evaluate the corresponding test accuracy. For all these three datasets, we sample 200 data points from their training dataset respectively as their individual validation dataset.

We follow the experimental setting of Lin et al. (2023): use the soft prompt to generate prompts from Vicuna-13B with a fixed intrinsic dimension of 1000 and search the soft token length {3, 5, 10} on the validation set. The corresponding prompt generation template is given below.

---

> **Prompt Generation Template for Chain-of-Thought**
>
> **I have some prompt examples for solving school math problems.**
>
> **prompt:**
> **Let's figure it out!**
>
> **prompt:**
> **Let's solve the problem.**
>
> **prompt:**
> **Let's think step by step.**
>
> **Write your new prompt that is different from the examples to solve the school math problems.**
>
> **prompt:**

See Tab. 2 below for the performances of our ZOPO against other baselines on the three arithmetic reasoning tasks. Our method achieves a better/comparable performance compared with other baselines.

Table 2: The performance of the best zero-shot CoT prompt found by different methods on three reasoning tasks.

| Method | Task | Best prompt | Score |
|---|---|---|---|
| hand-craft | AQUA-RAT | Let's think step by step. | 52.362 |
| InstructZero | AQUA-RAT | Let's break down the problem. | 54.331 |
| INSTINCT | AQUA-RAT | I have a new solution. | **54.724** |
| EvoPrompt | AQUA-RAT | Let's utilize the substitution method to find a solution, then try it out together. | 52.756 |
| ZOPO | AQUA-RAT | Let's find the solution by breaking down the problem. | **54.724** |
| hand-craft | SVAMP | Let's think step by step. | 76.25 |
| InstructZero | SVAMP | Let's use the equation. | 79.5 |
| INSTINCT | SVAMP | Let's use our brains. | **81.0** |
| EvoPrompt | SVAMP | Let's break down the issue at hand using promptal methods to gain a thorough analysis. | 79.5 |
| ZOPO | SVAMP | Let's use logic to solve the problem. | **81.0** |
| hand-craft | GSM8K | Let's think step by step. | 71.797 |
| InstructZero | GSM8K | Let's use the prompt to solve the problem. | 74.299 |
| INSTINCT | GSM8K | Let's think about it. | 74.526 |
| EvoPrompt | GSM8K | Let's attempt to analyze the situation and give it a shot. | 74.526 |
| ZOPO | GSM8K | Let's find the solution by using the given information. | **75.360** |

## C.5. Details on Compute Resources

All experiments are conducted on a server with Intel(R) Xeon(R) CPU and NVIDIA H100 GPUs. We mainly perform the prompt optimization for the GPT-3.5-Turbo model (for which OpenAI charges US0.5 per 1M tokens for input and US1.5 per 1M tokens for output). The time of execution of our algorithm on each prompt optimization task (e.g., any task in the 30 instruction induction tasks) normally takes less than 20 minutes, where the actual time would depend on OpenAI API's response speed.

# D. Additional Results

## D.1. Extended Empirical Study on Function Landscape

In Section 3, we have empirically studied the landscape of the target function and incorporated the findings into the design of ZOPO. In the main text, we have demonstrated the results on three instruction induction datasets, including *taxonomy_animal, cause_and_effect,* and *informal_to_formal*. Here we use more datasets to validate our findings. Due to the large size of instruction induction tasks (i.e., 30 tasks in total) and the query budget limit (i.e., it incurs monetary costs when we query the objective function ChatGPT to evaluate the prompt on the given task), we only experiment with few more randomly chosen tasks here to further validate our findings.

### D.1.1. LOCAL OPTIMA VS. GLOBAL OPTIMUM

To validate our local optimization design, we study the local optima in the function landscape, by using a 3-dimensional (reduced by t-SNE) scatter plot to represent the prompt embeddings (last token embeddings from Vicuna-13B). Here we provide the empirical results on more instruction induction tasks, shown in Fig. 6. The heatmap color represents the validation accuracy of the corresponding prompt. This allows us to interpret the local optima visually, and we conclude that many local optima can already exhibit compelling performance.

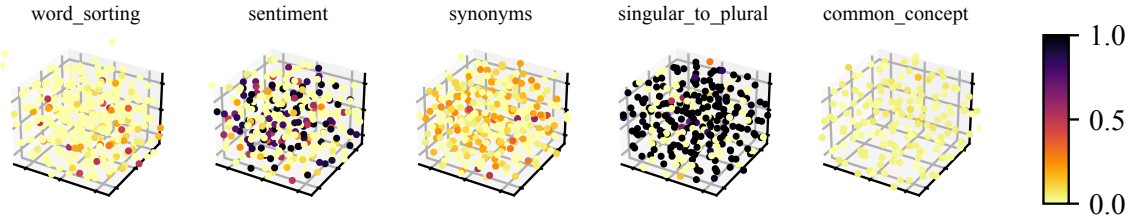

Figure 6: The validation accuracy of 300 randomly sampled prompts with the last token representation on various tasks.

### D.1.2. ESSENSE OF INPUT DOMAIN

**Prompt Generation**  To study the prompt quality of different prompt generation methods, we compare the prompts generated from Vicuna-13B and those generated from ChatGPT (i.e., GPT 3.5 turbo). For Vicuna-13B, we use the randomly sampled soft prompts with a fixed intrinsic dimension of 200 and a number token length of 10. For ChatGPT, we randomly sample prompts from the ChatGPT's response by using the APE generation template filled with random example demonstrations. For each generation method on each task, we generate 300 random prompts, and we query the target function with all prompts. We show the validation accuracy distribution of prompts generated by the two methods on four more (due to budget constraints) tasks here in Fig. 7. It demonstrates that ChatGPT has a larger probability of generating prompts with higher accuracy, also with a larger mean. The result shows that ChatGPT-generated prompts are generally better, further validating our finding of the importance of the input domain.

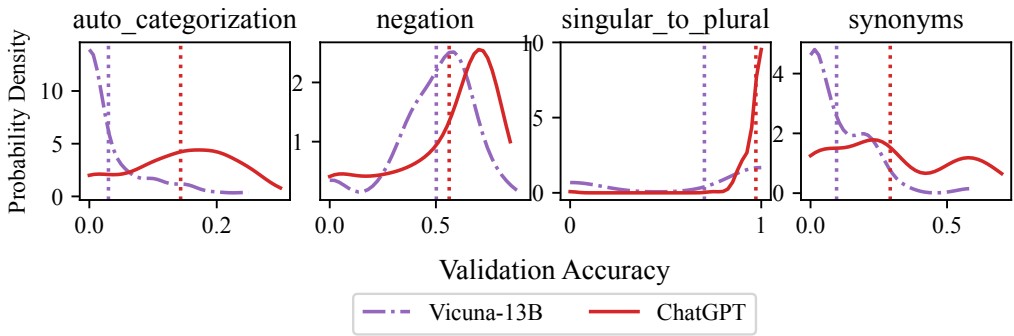

Figure 7: The estimated accuracy distribution of prompts generated by Vicuna-13B or ChatGPT on various instruction induction tasks, where the vertical dotted line indicates the mean performance.

**Prompt Embedding**   The complexity of modeling the target function depends on its function landscape defined by the embedding domain. To empirically analyze the black-box target function, we show the accuracy landscape of different tasks, where we reduce the dimension of the prompt embedding (we use the last token embedding of Vicuna-13B here) to 2 by using t-SNE. The loss landscape is visualized in the surface plot shown in Fig. 8. We observe that different optimization methods achieve similar performances on tasks like *sentiment* and *singular_to_plural*, as they have many good local optima. For other challenging tasks with complex function landscapes, the good local optima are less, but our methods can still achieve superior performance. This validates our insight that there are many good local optima in the embedding space.

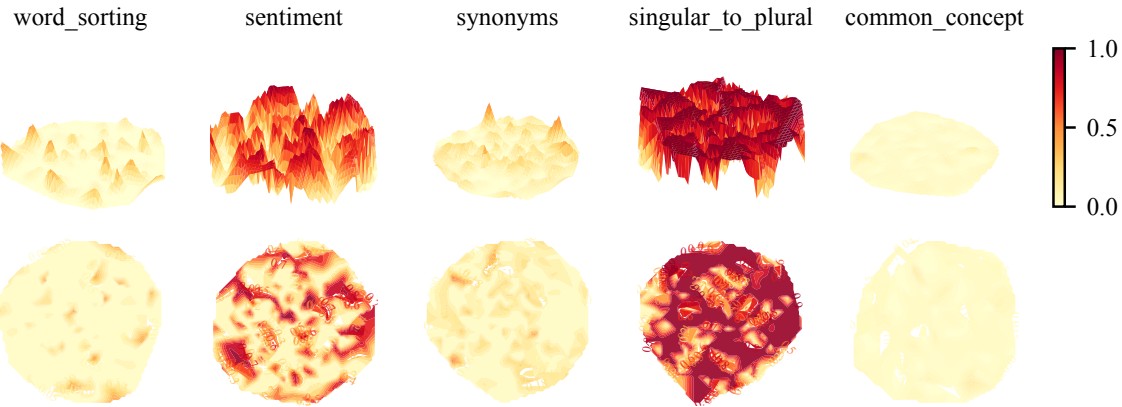

Figure 8: The function surfaces on various tasks using the last token embedding from Vicuna-13B as the representation for prompt candidates that are generated by Vicuna-13B, with contour plots shown below.

## D.2. Comparison on Instruction Induction Tasks

In Section 5.1 of the main text, we compared our methods with other baselines on 20 challenging instruction induction tasks. Here we provide the full results on 30 instruction induction tasks in Section 5.1.

Table 3: Average test accuracy with standard error (3 runs) for the best prompt found by different methods for all 30 instruction induction tasks.

| Tasks | APE | InstructZero | INSTINCT | EvoPrompt | PB | OPRO | ZOPO | ZOPO$_{\text{GPT}}$ |
|---|---|---|---|---|---|---|---|---|
| active_to_passive | **100.0**$_{\pm 0.0}$ | 99.7$_{\pm 0.3}$ | 97.0$_{\pm 2.5}$ | **100.0**$_{\pm 0.0}$ | 99.0$_{\pm 0.8}$ | **100.0**$_{\pm 0.0}$ | **100.0**$_{\pm 0.0}$ | **100.0**$_{\pm 0.0}$ |
| antonyms | 63.7$_{\pm 14.2}$ | 82.7$_{\pm 0.7}$ | 84.7$_{\pm 0.3}$ | 84.0$_{\pm 0.0}$ | 78.0$_{\pm 3.6}$ | 79.0$_{\pm 2.2}$ | **85.2**$_{\pm 3.2}$ | 84.0$_{\pm 1.4}$ |
| auto_categorization | 25.0$_{\pm 0.9}$ | 25.7$_{\pm 1.2}$ | 25.0$_{\pm 3.3}$ | 31.0$_{\pm 1.0}$ | 24.0$_{\pm 3.7}$ | 24.0$_{\pm 3.6}$ | **32.7**$_{\pm 1.9}$ | 27.0$_{\pm 5.0}$ |
| auto_debugging | 29.2$_{\pm 3.4}$ | 37.5$_{\pm 0.0}$ | 29.2$_{\pm 3.4}$ | 33.0$_{\pm 7.2}$ | 25.0$_{\pm 0.0}$ | 37.5$_{\pm 0.0}$ | **41.7**$_{\pm 15.6}$ | 29.2$_{\pm 5.9}$ |
| cause_and_effect | 57.3$_{\pm 8.9}$ | 81.33$_{\pm 1.1}$ | 58.7$_{\pm 8.7}$ | 84.0$_{\pm 13.9}$ | 82.7$_{\pm 10.0}$ | 82.7$_{\pm 10.0}$ | **94.7**$_{\pm 3.7}$ | 80.0$_{\pm 14.2}$ |
| common_concept | 6.9$_{\pm 2.1}$ | 8.6$_{\pm 4.0}$ | 21.3$_{\pm 0.2}$ | 11.1$_{\pm 6.9}$ | 10.9$_{\pm 3.4}$ | 8.6$_{\pm 5.7}$ | **23.5**$_{\pm 3.4}$ | 2.8$_{\pm 0.6}$ |
| diff | 67.3$_{\pm 26.7}$ | 69.3$_{\pm 22.2}$ | **100.0**$_{\pm 0.0}$ | 27.3$_{\pm 42.2}$ | 71.3$_{\pm 27.6}$ | **100.0**$_{\pm 0.0}$ | **100.0**$_{\pm 0.0}$ | **100.0**$_{\pm 0.0}$ |
| first_word_letter | **100.0**$_{\pm 0.0}$ | **100.0**$_{\pm 0.0}$ | 93.0$_{\pm 5.3}$ | **100.0**$_{\pm 0.0}$ | **100.0**$_{\pm 0.0}$ | **100.0**$_{\pm 0.0}$ | **100.0**$_{\pm 0.0}$ | **100.0**$_{\pm 0.0}$ |
| informal_to_formal | 57.4$_{\pm 0.3}$ | 53.1$_{\pm 0.2}$ | 55.3$_{\pm 0.0}$ | 51.6$_{\pm 0.9}$ | 54.2$_{\pm 4.5}$ | 48.0$_{\pm 0.8}$ | 61.3$_{\pm 2.7}$ | **61.9**$_{\pm 2.9}$ |
| larger_animal | 89.7$_{\pm 0.5}$ | 90.0$_{\pm 4.1}$ | **93.7**$_{\pm 0.3}$ | 87.3$_{\pm 3.1}$ | 73.3$_{\pm 9.1}$ | 90.7$_{\pm 4.1}$ | 92.3$_{\pm 2.9}$ | 92.7$_{\pm 1.2}$ |
| letters_list | **100.0**$_{\pm 0.0}$ | 59.0$_{\pm 16.7}$ | **100.0**$_{\pm 0.0}$ | **100.0**$_{\pm 0.0}$ | 99.3$_{\pm 0.9}$ | 99.7$_{\pm 0.5}$ | **100.0**$_{\pm 0.0}$ | **100.0**$_{\pm 0.0}$ |
| negation | 75.3$_{\pm 1.1}$ | 77.7$_{\pm 1.4}$ | 81.7$_{\pm 0.3}$ | 86.0$_{\pm 0.0}$ | 70.7$_{\pm 4.0}$ | 73.3$_{\pm 6.6}$ | **86.3**$_{\pm 0.5}$ | 77.7$_{\pm 2.6}$ |
| num_to_verbal | 99.7$_{\pm 0.3}$ | **100.0**$_{\pm 0.0}$ | **100.0**$_{\pm 0.0}$ | **100.0**$_{\pm 0.0}$ | 98.3$_{\pm 1.7}$ | **100.0**$_{\pm 0.0}$ | **100.0**$_{\pm 0.0}$ | **100.0**$_{\pm 0.0}$ |
| object_counting | 36.3$_{\pm 1.9}$ | 36.0$_{\pm 9.3}$ | 34.0$_{\pm 7.0}$ | **55.0**$_{\pm 5.3}$ | 29.3$_{\pm 1.2}$ | 36.0$_{\pm 5.7}$ | 52.3$_{\pm 6.6}$ | 40.3$_{\pm 0.5}$ |
| odd_one_out | 63.3$_{\pm 1.4}$ | 61.3$_{\pm 8.7}$ | **70.0**$_{\pm 1.6}$ | 10.0$_{\pm 0.0}$ | 66.7$_{\pm 0.9}$ | 47.3$_{\pm 22.2}$ | 32.0$_{\pm 11.3}$ | 68.7$_{\pm 2.5}$ |
| orthography_starts_with | 45.7$_{\pm 14.8}$ | 50.7$_{\pm 8.7}$ | 66.7$_{\pm 2.7}$ | 15.0$_{\pm 3.4}$ | 59.8$_{\pm 5.7}$ | 33.5$_{\pm 14.6}$ | 56.5$_{\pm 12.6}$ | **71.0**$_{\pm 0.0}$ |
| periodic_elements | 92.7$_{\pm 2.2}$ | 86.7$_{\pm 6.1}$ | 92.7$_{\pm 2.7}$ | 98.0$_{\pm 1.2}$ | 95.3$_{\pm 2.5}$ | 93.3$_{\pm 0.9}$ | **100.0**$_{\pm 0.0}$ | 94.7$_{\pm 3.1}$ |
| rhymes | 15.7$_{\pm 6.4}$ | **100.0**$_{\pm 0.0}$ | **100.0**$_{\pm 0.0}$ | 59.7$_{\pm 3.1}$ | 45.0$_{\pm 10.7}$ | 23.0$_{\pm 14.7}$ | **100.0**$_{\pm 0.0}$ | 61.0$_{\pm 2.8}$ |
| second_word_letter | 74.7$_{\pm 20.3}$ | 43.3$_{\pm 18.7}$ | 10.0$_{\pm 4.1}$ | 24.7$_{\pm 0.6}$ | 88.7$_{\pm 10.4}$ | 86.7$_{\pm 18.9}$ | 25.7$_{\pm 4.7}$ | **96.7**$_{\pm 2.4}$ |
| sentence_similarity | 0.0$_{\pm 0.0}$ | 0.0$_{\pm 0.0}$ | 14.0$_{\pm 0.5}$ | 2.0$_{\pm 1.0}$ | 0.0$_{\pm 0.0}$ | 2.7$_{\pm 3.8}$ | 7.6$_{\pm 9.3}$ | **37.3**$_{\pm 0.9}$ |
| sentiment | 91.3$_{\pm 1.4}$ | 87.7$_{\pm 2.4}$ | 89.7$_{\pm 1.4}$ | 93.0$_{\pm 0.0}$ | 81.7$_{\pm 6.0}$ | 57.2$_{\pm 39.9}$ | **93.5**$_{\pm 0.5}$ | 89.3$_{\pm 2.1}$ |
| singular_to_plural | **100.0**$_{\pm 0.0}$ | 98.7$_{\pm 1.1}$ | **100.0**$_{\pm 0.0}$ | **100.0**$_{\pm 0.0}$ | 98.0$_{\pm 0.8}$ | **100.0**$_{\pm 0.0}$ | **100.0**$_{\pm 0.0}$ | **100.0**$_{\pm 0.0}$ |
| sum | 67.3$_{\pm 26.7}$ | **100.0**$_{\pm 0.0}$ | **100.0**$_{\pm 0.0}$ | **100.0**$_{\pm 0.0}$ | 98.3$_{\pm 1.7}$ | **100.0**$_{\pm 0.0}$ | **100.0**$_{\pm 0.0}$ | **100.0**$_{\pm 0.0}$ |
| synonyms | 36.0$_{\pm 7.6}$ | 27.7$_{\pm 9.3}$ | 30.7$_{\pm 4.9}$ | 40.3$_{\pm 4.0}$ | 36.3$_{\pm 3.3}$ | 40.0$_{\pm 4.3}$ | 43.3$_{\pm 0.9}$ | **44.7**$_{\pm 4.1}$ |
| taxonomy_animal | 34.7$_{\pm 23.4}$ | 71.7$_{\pm 8.4}$ | 85.7$_{\pm 6.0}$ | 83.0$_{\pm 4.6}$ | 29.7$_{\pm 38.5}$ | 30.0$_{\pm 41.0}$ | 90.0$_{\pm 7.1}$ | **92.3**$_{\pm 0.5}$ |
| translation_en-de | 84.0$_{\pm 0.5}$ | 82.3$_{\pm 0.1}$ | 84.0$_{\pm 0.5}$ | 85.0$_{\pm 0.0}$ | 71.3$_{\pm 11.9}$ | 79.7$_{\pm 4.0}$ | **85.3**$_{\pm 0.5}$ | 84.7$_{\pm 0.6}$ |
| translation_en-es | 87.0$_{\pm 0.0}$ | 87.3$_{\pm 0.1}$ | **88.0**$_{\pm 0.0}$ | 82.3$_{\pm 7.4}$ | 81.7$_{\pm 1.7}$ | 85.3$_{\pm 2.4}$ | 85.3$_{\pm 2.1}$ | 86.3$_{\pm 2.5}$ |
| translation_en-fr | 88.7$_{\pm 0.3}$ | 87.7$_{\pm 0.0}$ | 83.0$_{\pm 2.1}$ | 80.7$_{\pm 4.5}$ | 76.7$_{\pm 8.0}$ | 86.0$_{\pm 2.2}$ | **91.0**$_{\pm 0.0}$ | 86.7$_{\pm 2.1}$ |
| word_sorting | 33.0$_{\pm 3.7}$ | 31.0$_{\pm 11.4}$ | 51.3$_{\pm 0.3}$ | 48.0$_{\pm 21.3}$ | 45.7$_{\pm 1.7}$ | 50.3$_{\pm 5.8}$ | 60.0$_{\pm 4.2}$ | **60.3**$_{\pm 3.1}$ |
| word_unscrambling | 44.0$_{\pm 13.9}$ | 59.0$_{\pm 5.3}$ | **63.3**$_{\pm 0.7}$ | 51.3$_{\pm 4.5}$ | 51.0$_{\pm 6.2}$ | 61.3$_{\pm 2.1}$ | 59.3$_{\pm 2.8}$ | 58.3$_{\pm 1.9}$ |
| *# best-performing tasks* | 4 | 4 | 10 | 7 | 1 | 6 | **18** | 15 |
| performance profile $\rho(5)$ | 0.37 | 0.43 | 0.57 | 0.47 | 0.27 | 0.43 | **0.87** | 0.73 |

The performance profile of ZOPO_GPT compared against other baseline methods is shown in Fig. 9. This corresponds to the result shown in Tab. 1.

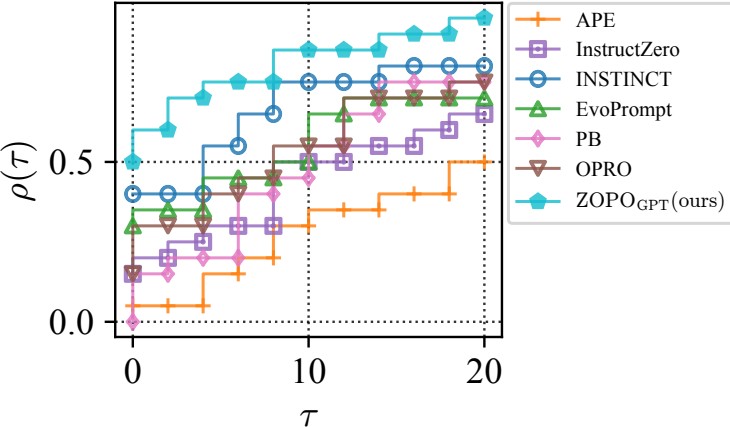

Figure 9: The performance profile of ZOPO_GPT compared against other baseline methods on 20 instruction induction tasks.

We also provide additional results on other instruction induction tasks to compare ZOPO against baseline methods in terms of query efficiency. The result is shown in Fig. 10.

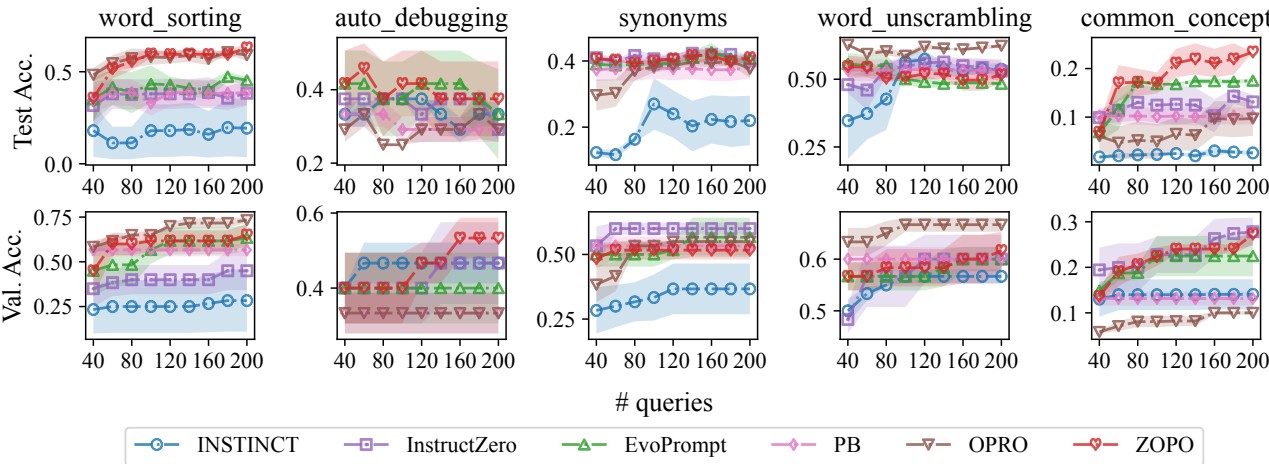

Figure 10: Comparison of the query efficiency between ZOPO and other existing baselines on various instruction induction tasks.

### D.3. Results on the GLUE Benchmark

We here follow the same experimental setting as our previous experiments on the instruction induction tasks and apply the prompt optimization on the GLUE benchmark, which consists of several more traditional natural language processing tasks. The result in Tab. 4 shows that our method ZOPO is still able to achieve advanced performance when compared with other baselines.

Table 4: Test accuracy achieved by different methods on GLUE tasks.

| Tasks | APE | InstructZero | INSTINCT | EvoPrompt | PromptBeeder | OPRO | ZOPO |
|---|---|---|---|---|---|---|---|
| CoLA | **82.0** | 79.0 | 72.0 | 65.0 | 54.0 | 0.0 | 65.0 |
| MNLI-m | **72.0** | 70.0 | 66.0 | **72.0** | 69.0 | 71.0 | 70.0 |
| MNLI-mm | 71.0 | 40.0 | 66.0 | 64.0 | 56.0 | 52.0 | **73.0** |
| MRPC | 66.0 | **76.0** | 71.0 | 71.0 | 28.0 | 0.0 | **76.0** |
| QNLI | 80.0 | 83.0 | 78.0 | 77.0 | 83.0 | **84.0** | 83.0 |
| QQP | 78.0 | 79.0 | **83.0** | 43.0 | 72.0 | 75.0 | **83.0** |
| RTE | 83.0 | **86.0** | 74.0 | 76.0 | 20.0 | 79.0 | 83.0 |
| SST-2 | 92.0 | **97.0** | 93.0 | 93.0 | 91.0 | 85.0 | 92.0 |
| # *best-performing tasks* | 2 | **3** | 1 | 1 | 0 | 1 | **3** |
| performance profile $\rho(5)$ | **0.875** | **0.875** | 0.375 | 0.375 | 0.25 | 0.25 | **0.875** |

### D.4. Few-shot Setting

Our algorithm ZOPO is in fact able to handle the few-shot settings as evidenced by the results in Tab. 5 below. Interestingly, the performance of our ZOPO is even better in a few-shot setting, which is reasonable since in-context exemplars will help the black-box models better understand the context and the output format of the task, and consequently will be able to lead to a better performance than the zero-shot setting. In this few-shot experiment, we provide exemplars for prompt evaluation and also report the test accuracy of the best prompt with exemplars provided.

Table 5: Test accuracy achieved by ZOPO under zero-shot and few-shot settings on instruction induction tasks.

| Tasks | Zero-shot | Few-shot (5) |
|---|---|---|
| antonyms | 81.0 | **86.0** |
| auto_categorization | **38.0** | 34.0 |
| auto_debugging | 37.5 | **50.0** |
| cause_and_effect | **92.0** | **92.0** |
| common_concept | **19.6** | 10.2 |
| diff | 97.0 | **99.0** |
| informal_to_formal | **63.4** | 46.9 |
| letters_list | **100.0** | **100.0** |
| negation | 86.0 | **90.0** |
| object_counting | **57.0** | 55.0 |
| odd_one_out | 6.0 | **36.0** |
| orthography_starts_with | 44.0 | **67.0** |
| rhymes | **98.0** | 65.0 |
| second_word_letter | 16.0 | **65.0** |
| sentence_similarity | 18.0 | **27.0** |
| sum | **100.0** | **100.0** |
| synonyms | **44.0** | 25.0 |
| taxonomy_animal | **97.0** | 62.0 |
| word_sorting | 57.0 | **62.0** |
| word_unscrambling | 61.0 | **62.0** |
| *# best-performing tasks* | 10 | **13** |

## D.5. Results of Different Combinations of Generation and Evaluation Models

We use further experiments to show that our method can generalize to different prompt generation models and different black-box API LLMs for prompt evaluation (as $f(\cdot)$). Specifically, we here consider two open-sourced models: Vicuna-13B and WizardLM-13B (Xu et al., 2024) for the prompt generation and we use their corresponding last token embeddings in our algorithm. For the black-box API LLMs, we consider GPT-3.5 (the one considered in our main text), PaLM2 (Anil et al., 2023), and GPT-4. In total, we have six combinations. Tab. 6 shows the results of different combinations on the instruction induction tasks. The results show that our ZOPO performs well on all these black-box API models (with GPT-4 performing the best on most tasks), which further verifies the generality of our method when a different black-box LLM $f(\cdot)$ is considered in the objective function in Eq. 1. We also notice that the Vicuna model generally performs better than the WizardLM model, which suggests it is more suitable for prompt generation and representation when applying our method ZOPO.

Table 6: Test accuracy on instruction induction tasks with different black-box LLMs $f(\cdot)$ considered in the objective function in Eq. 1.

| Prompt Generation/Embedding LLM | Vicuna | Vicuna | Vicuna | WizardLM | WizardLM | WizardLM |
|---|---|---|---|---|---|---|
| **Black-box LLM (Objective Function)** | GPT-3.5 | PaLM2 | GPT-4 | GPT-3.5 | PaLM2 | GPT-4 |
| antonyms | 81.0 | 80.0 | 85.0 | 80.0 | 80.0 | **88.0** |
| auto_categorization | **38.0** | 24.3 | 37.0 | 19.0 | 1.0 | 3.0 |
| auto_debugging | **37.5** | 33.3 | 25.0 | 25.0 | **37.5** | **37.5** |
| cause_and_effect | 92.0 | **96.0** | **96.0** | 48.0 | 76.0 | 68.0 |
| common_concept | **19.6** | 7.9 | 19.3 | 14.8 | 18.2 | 13.5 |
| diff | 97.0 | **100.0** | **100.0** | 99.0 | **100.0** | **100.0** |
| informal_to_formal | **63.4** | 50.1 | 51.5 | 44.8 | 53.1 | 41.9 |
| letters_list | **100.0** | 94.0 | **100.0** | **100.0** | 95.0 | **100.0** |
| negation | **86.0** | 80.7 | 84.0 | 73.0 | 79.0 | 73.0 |
| object_counting | 57.0 | 57.3 | 57.0 | 49.0 | 48.0 | **59.0** |
| odd_one_out | 6.0 | 14.7 | **68.0** | 36.0 | 24.0 | 26.0 |
| orthography_starts_with | 44.0 | 60.0 | **72.0** | 34.0 | 30.0 | 13.0 |
| rhymes | 98.0 | 84.0 | 86.0 | 53.0 | 90.0 | **100.0** |
| second_word_letter | 16.0 | 22.0 | **99.0** | 1.0 | 18.0 | 93.0 |
| sentence_similarity | **18.0** | 0.0 | 5.0 | 11.0 | 0.0 | 0.0 |
| sum | **100.0** | 72.0 | **100.0** | 98.0 | **100.0** | **100.0** |
| synonyms | **44.0** | 35.3 | 41.0 | 36.0 | 16.0 | 39.0 |
| taxonomy_animal | 97.0 | 86.3 | **100.0** | 95.0 | 77.0 | 97.0 |
| word_sorting | 57.0 | 19.3 | **66.0** | 0.0 | 1.0 | 0.0 |
| word_unscrambling | 61.0 | 12.7 | **71.0** | 51.0 | 14.0 | 58.0 |
| # *best-performing tasks* | 9 | 2 | **10** | 1 | 3 | 7 |
| performance profile $\rho(5)$ | 0.7 | 0.2 | **0.8** | 0.25 | 0.25 | 0.45 |

## D.6. Verifying the Essence of Input Domain

**Prompt Generation**   To fairly compare the effect of prompts generated by Vicuna-13B and ChatGPT in terms of the optimization performance by using ZOPO, we adopt the same embedding representations here, that is we use the SBERT embedding model for both prompts generated by Vicuna-13B and ChatGPT. For the prompt generation process, we fix the number of prompt candidates for both methods to 500. The result of the comparison on 20 instruction induction tasks is shown in Table. 7, where the corresponding performance profile shown in Fig. 11 suggests that applying ZOPO on ChatGPT-generated prompts is better than applying it on Vicuna-generated prompts. This again confirms the importance of the choice of the input domain (i.e., the prompt generation).

Table 7: Fair comparison of the optimization performance of ZOPO with different generated prompts but the same embedding model (i.e., SBERT).

Figure 11: The corresponding performance profile for results shown in Tab. 7.

| Tasks | Vicuna-13B | ChatGPT |
|---|---|---|
| antonyms | $78.3_{\pm 4.5}$ | $\mathbf{84.0}_{\pm 1.4}$ |
| auto_categorization | $\mathbf{29.7}_{\pm 2.9}$ | $27.0_{\pm 5.0}$ |
| auto_debugging | $\mathbf{41.7}_{\pm 15.6}$ | $29.2_{\pm 5.9}$ |
| cause_and_effect | $\mathbf{86.7}_{\pm 7.5}$ | $80.0_{\pm 14.2}$ |
| common_concept | $\mathbf{24.9}_{\pm 0.0}$ | $2.8_{\pm 0.6}$ |
| diff | $8.0_{\pm 7.1}$ | $\mathbf{100.0}_{\pm 0.0}$ |
| informal_to_formal | $\mathbf{62.0}_{\pm 3.3}$ | $61.9_{\pm 2.9}$ |
| letters_list | $\mathbf{100.0}_{\pm 0.0}$ | $\mathbf{100.0}_{\pm 0.0}$ |
| negation | $\mathbf{82.0}_{\pm 2.9}$ | $77.7_{\pm 2.6}$ |
| object_counting | $\mathbf{45.3}_{\pm 10.3}$ | $40.3_{\pm 0.5}$ |
| odd_one_out | $20.0_{\pm 3.3}$ | $\mathbf{68.7}_{\pm 2.5}$ |
| orthography_starts_with | $51.0_{\pm 6.1}$ | $\mathbf{71.0}_{\pm 0.0}$ |
| rhymes | $\mathbf{100.0}_{\pm 0.0}$ | $61.0_{\pm 2.8}$ |
| second_word_letter | $24.3_{\pm 6.0}$ | $\mathbf{96.7}_{\pm 2.4}$ |
| sentence_similarity | $10.3_{\pm 14.6}$ | $\mathbf{37.3}_{\pm 0.9}$ |
| sum | $\mathbf{100.0}_{\pm 0.0}$ | $\mathbf{100.0}_{\pm 0.0}$ |
| synonyms | $40.3_{\pm 1.7}$ | $\mathbf{44.7}_{\pm 4.1}$ |
| taxonomy_animal | $91.7_{\pm 2.1}$ | $\mathbf{92.3}_{\pm 0.5}$ |
| word_sorting | $\mathbf{62.7}_{\pm 0.5}$ | $60.3_{\pm 3.1}$ |
| word_unscrambling | $53.0_{\pm 0.0}$ | $\mathbf{58.3}_{\pm 1.9}$ |

**Prompt Embedding** Here we analyze how different embeddings affect the optimization of ZOPO. We first generate a fixed set of prompts of size 500 from Vicuna-13B as those in Tab. 1. For the same set of prompts, we consider four different embeddings here: (a) the **Last Token** embedding from Vicuna-13B (b) the **OpenAI** embedding obtained through its embedding model "text-embedding-ada-002" API. (OpenAI, 2024b), (c) the **SBERT** embedding obtained through the sentence transformer ("all-mpnet-base-v2" from HuggingFace), and (d) the **Random** embedding obtained by randomly projecting the Vicuna embedding into the same dimension. The dimensions of the four embeddings (from (a) to (d)) are 1536, 756, and 5120 respectively. We compare the optimization performance of the four embeddings using ZOPO and the results are shown in Tab. 8. We observe although last token embedding is generally better, there are certain tasks that OpenAI and SBERT embeddings perform equally well or better, which indicates the optimal choice of embedding can be *task-dependent*. Intuitively, random embedding is not representative. Its lesser performance shown in Tab. 8 again confirms our Insight II in Sec. 3.2, which says the choice of embedding/input domain is important in prompt optimization.

Table 8: Average test accuracy with standard error (3 runs) for the best prompt found by ZOPO with four different embeddings on 20 instruction induction tasks.

| Tasks | Last Token (5120) | OpenAI (1536) | SBERT (756) | Random (5120) |
|---|---|---|---|---|
| antonyms | $\mathbf{85.2}_{\pm 3.2}$ | $76.7_{\pm 0.4}$ | $78.3_{\pm 4.5}$ | $79.3_{\pm 3.4}$ |
| auto_categorization | $\mathbf{32.7}_{\pm 1.9}$ | $31.0_{\pm 2.9}$ | $29.7_{\pm 2.9}$ | $32.3_{\pm 1.7}$ |
| auto_debugging | $\mathbf{41.7}_{\pm 15.6}$ | $29.2_{\pm 5.9}$ | $\mathbf{41.7}_{\pm 15.6}$ | $37.5_{\pm 17.7}$ |
| cause_and_effect | $\mathbf{94.7}_{\pm 3.7}$ | $82.7_{\pm 6.8}$ | $86.7_{\pm 7.5}$ | $68.0_{\pm 8.6}$ |
| common_concept | $23.5_{\pm 3.4}$ | $24.4_{\pm 1.5}$ | $\mathbf{24.9}_{\pm 0.0}$ | $22.4_{\pm 1.8}$ |
| diff | $\mathbf{100.0}_{\pm 0.0}$ | $94.7_{\pm 3.1}$ | $8.0_{\pm 7.1}$ | $15.7_{\pm 7.4}$ |
| informal_to_formal | $61.3_{\pm 2.7}$ | $59.4_{\pm 2.4}$ | $\mathbf{62.0}_{\pm 3.3}$ | $58.5_{\pm 3.7}$ |
| letters_list | $\mathbf{100.0}_{\pm 0.0}$ | $\mathbf{100.0}_{\pm 0.0}$ | $\mathbf{100.0}_{\pm 0.0}$ | $\mathbf{100.0}_{\pm 0.0}$ |
| negation | $\mathbf{86.3}_{\pm 0.5}$ | $82.3_{\pm 1.9}$ | $82.0_{\pm 2.9}$ | $84.0_{\pm 2.2}$ |
| object_counting | $\mathbf{52.3}_{\pm 6.6}$ | $51.7_{\pm 6.1}$ | $45.3_{\pm 10.3}$ | $51.7_{\pm 6.2}$ |
| odd_one_out | $\mathbf{32.0}_{\pm 11.3}$ | $24.0_{\pm 8.6}$ | $20.0_{\pm 3.3}$ | $20.0_{\pm 12.3}$ |
| orthography_starts_with | $\mathbf{56.5}_{\pm 12.6}$ | $56.0_{\pm 4.3}$ | $51.0_{\pm 6.1}$ | $46.7_{\pm 4.7}$ |
| rhymes | $\mathbf{100.0}_{\pm 0.0}$ | $68.7_{\pm 21.5}$ | $\mathbf{100.0}_{\pm 0.0}$ | $96.3_{\pm 2.4}$ |
| second_word_letter | $\mathbf{25.7}_{\pm 4.7}$ | $24.3_{\pm 5.2}$ | $24.3_{\pm 6.0}$ | $24.3_{\pm 4.5}$ |
| sentence_similarity | $7.6_{\pm 9.3}$ | $\mathbf{10.3}_{\pm 14.6}$ | $\mathbf{10.3}_{\pm 14.6}$ | $6.3_{\pm 6.4}$ |
| sum | $\mathbf{100.0}_{\pm 0.0}$ | $\mathbf{100.0}_{\pm 0.0}$ | $\mathbf{100.0}_{\pm 0.0}$ | $\mathbf{100.0}_{\pm 0.0}$ |
| synonyms | $\mathbf{43.3}_{\pm 0.9}$ | $40.0_{\pm 0.0}$ | $40.3_{\pm 1.7}$ | $42.3_{\pm 3.1}$ |
| taxonomy_animal | $90.0_{\pm 7.1}$ | $\mathbf{91.7}_{\pm 2.6}$ | $\mathbf{91.7}_{\pm 2.1}$ | $89.3_{\pm 6.2}$ |
| word_sorting | $60.0_{\pm 4.2}$ | $\mathbf{63.0}_{\pm 1.4}$ | $62.7_{\pm 0.5}$ | $59.7_{\pm 3.8}$ |
| word_unscrambling | $\mathbf{59.3}_{\pm 2.8}$ | $56.3_{\pm 1.7}$ | $53.0_{\pm 0.0}$ | $47.3_{\pm 4.2}$ |
| *# best-performing tasks* | **15** | 5 | 8 | 2 |

To demonstrate the generality of ZOPO, we provide more results below to show we are not focusing on a specific combination of LLMs and embedding models. For the same optimization objective (i.e., we still perform prompt optimization for ChatGPT), we further study our ZOPO with more choices of embedding models and pair it with different black-box LLMs for prompt generation (i.e., GPT-4). We consider three more embedding models from HuggingFace and OpenAI, including "Instructor-Large", "MiniLM-L6-v2", and "text-embedding-3-small" in Tab. 9 below, as well another black-box model (i.e., GPT-4) for prompt generation in Tab. 10 below.

Note that, in our main text, we use the Vicuna-13B model as the prompt generation model mainly for the fair comparison against baselines (i.e., InstructZero and INSTINCT), and we can tell from Tab. 9 that the Vicuna embedding is a good embedding to use in prompt optimization. The result from Tab. 10 suggests choosing a better embedding model (i.e.,

Instructor-Large) for GPT-4 generated prompts can even further improve its performance.

**Can we possibly find the best embedding model?**    As we have shown in our previous experiments, the best choice for the embedding model can be *task-dependent*. To find the suitable paired embedding model without enumerating every embedding model, we could possibly analyze the variance $V_e$ of the eigenvalues of the covariance matrix of the embeddings (i.e., $\mathcal{Z}$). The eigenvalues represent the variances along the principal directions (eigenvectors) of the embeddings. If the embeddings are distributed with equal spacing in the high-dimensional space, we would expect the eigenvalues to be approximately equal (i.e., low variance). Intuitively, if the embeddings are in such representative space with equal spacing, it could help function modeling.

Therefore, if such variance $V_e$ is small, the optimization performance using such an embedding model is more likely to be better. Based on the test accuracies from Tab. 9 and Tab. 10, we show that there exists a sufficiently high negative Spearsman's correlation (i.e., the average correlation is -0.47) between $V_e$ and the performance of ZOPO using different embedding models on each task. Therefore, we can check every embedding prior to the experiment, and it can be done efficiently. We acknowledge that this approach is not perfect and finding the best embedding model is not the main focus of this work. By finding the best pair of generation and embedding models for prompt optimization, we believe our ZOPO algorithm can be further boosted. We would like to take it as a potential future direction.

Table 9: Test accuracy achieved by ZOPO (Vicuna-13B for prompt generation) with different embeddings on GLUE tasks.

| **Tasks** | Vicuna | Instructor-Large | MiniLM-L6-v2 | text-embedding-3-small |
|---|---|---|---|---|
| CoLA | 65.0 | **70.0** | 58.0 | 60.0 |
| MNLI-m | 70.0 | 67.0 | **71.0** | **71.0** |
| MNLI-mm | **73.0** | 62.0 | **73.0** | 62.0 |
| MRPC | **76.0** | 73.0 | 73.0 | 73.0 |
| QNLI | **83.0** | 78.0 | **83.0** | **83.0** |
| QQP | **83.0** | 82.0 | 82.0 | 82.0 |
| RTE | **83.0** | 78.0 | 75.0 | 82.0 |
| SST-2 | **92.0** | 91.0 | 91.0 | 91.0 |
| *# best-performing tasks* | **6** | 1 | 3 | 2 |
| performance profile $\rho(5)$ | **1.00** | 0.88 | 0.75 | 0.75 |

Table 10: Test accuracy achieved by ZOPO$_{\text{GPT-4}}$ (GPT-4 for prompt generation) with different embeddings on GLUE tasks

| **Tasks** | SBERT | Instructor-Large | MiniLM-L6-v2 | text-embedding-3-small |
|---|---|---|---|---|
| CoLA | 77.0 | 78.0 | **79.0** | 76.0 |
| MNLI-m | 73.0 | 73.0 | **74.0** | 69.0 |
| MNLI-mm | 72.0 | **77.0** | **77.0** | 69.0 |
| MRPC | 68.0 | **71.0** | 65.0 | **71.0** |
| QNLI | 83.0 | 83.0 | **84.0** | 83.0 |
| QQP | **81.0** | 76.0 | 74.0 | 75.0 |
| RTE | 84.0 | **86.0** | 81.0 | 82.0 |
| SST-2 | **96.0** | **96.0** | 93.0 | 93.0 |
| *# best-performing tasks* | 2 | **4** | **4** | 1 |
| performance profile $\rho(5)$ | **1.0** | **1.0** | 0.75 | 0.75 |

## D.7. Study of NTK-GP and Uncertainty-Informed Local Exploration

To validate the effectiveness of the components, namely NTK-GP (in Sec. 4.2) and uncertainty- informed local exploration (in Sec. 4.3) of ZOPO, we perform controlled experiments to replace these components. Specifically, we (a) replace the NTK component with Matérn kernel (as in the recent ZOO method ZoRD), and (b) remove the uncertainty-informed local exploration feature. We evaluate the two settings on 20 instruction induction tasks. The result shown in Table 11 illustrates these two settings are both significantly worse than the original ZOPO, which validates the effectiveness of NTK-GP and uncertainty-informed local exploration.

Table 11: Ablation study of the design components in ZOPO showing the average test accuracy reported with standard error (3 runs) on 20 instruction induction tasks.

| Tasks | ZOPO | ZOPO w/o NTK | ZOPO w/o Local Exploration |
|---|---|---|---|
| antonyms | $\mathbf{85.2}_{\pm 3.2}$ | $79.7_{\pm 9.0}$ | $78.7_{\pm 3.1}$ |
| auto_categorization | $32.7_{\pm 1.9}$ | $\mathbf{34.7}_{\pm 3.7}$ | $28.3_{\pm 4.9}$ |
| auto_debugging | $\mathbf{41.7}_{\pm 15.6}$ | $29.2_{\pm 5.9}$ | $25.0_{\pm 0.0}$ |
| cause_and_effect | $\mathbf{94.7}_{\pm 3.7}$ | $93.3_{\pm 1.9}$ | $85.3_{\pm 6.8}$ |
| common_concept | $\mathbf{23.5}_{\pm 3.4}$ | $9.2_{\pm 4.1}$ | $22.0_{\pm 5.6}$ |
| diff | $\mathbf{100.0}_{\pm 0.0}$ | $13.7_{\pm 6.1}$ | $13.7_{\pm 6.1}$ |
| informal_to_formal | $61.3_{\pm 2.7}$ | $\mathbf{63.4}_{\pm 0.0}$ | $\mathbf{63.4}_{\pm 0.0}$ |
| letters_list | $\mathbf{100.0}_{\pm 0.0}$ | $\mathbf{100.0}_{\pm 0.0}$ | $\mathbf{100.0}_{\pm 0.0}$ |
| negation | $\mathbf{86.3}_{\pm 0.5}$ | $85.7_{\pm 0.5}$ | $84.7_{\pm 3.3}$ |
| object_counting | $\mathbf{52.3}_{\pm 6.6}$ | $39.0_{\pm 7.1}$ | $51.7_{\pm 6.2}$ |
| odd_one_out | $\mathbf{32.0}_{\pm 11.3}$ | $14.7_{\pm 5.0}$ | $\mathbf{32.0}_{\pm 8.6}$ |
| orthography_starts_with | $\mathbf{56.5}_{\pm 12.6}$ | $49.3_{\pm 8.2}$ | $46.3_{\pm 9.7}$ |
| rhymes | $\mathbf{100.0}_{\pm 0.0}$ | $90.7_{\pm 0.5}$ | $93.3_{\pm 6.6}$ |
| second_word_letter | $\mathbf{25.7}_{\pm 4.7}$ | $\mathbf{25.7}_{\pm 6.8}$ | $19.7_{\pm 6.8}$ |
| sentence_similarity | $\mathbf{7.6}_{\pm 9.3}$ | $0.0_{\pm 0.0}$ | $0.0_{\pm 0.0}$ |
| sum | $\mathbf{100.0}_{\pm 0.0}$ | $93.7_{\pm 9.0}$ | $\mathbf{100.0}_{\pm 0.0}$ |
| synonyms | $\mathbf{43.3}_{\pm 0.9}$ | $38.3_{\pm 0.9}$ | $39.7_{\pm 2.5}$ |
| taxonomy_animal | $90.0_{\pm 7.1}$ | $74.7_{\pm 15.1}$ | $\mathbf{91.3}_{\pm 4.1}$ |
| word_sorting | $\mathbf{60.0}_{\pm 4.2}$ | $29.3_{\pm 12.7}$ | $56.3_{\pm 0.9}$ |
| word_unscrambling | $\mathbf{59.3}_{\pm 2.8}$ | $47.3_{\pm 0.9}$ | $50.0_{\pm 4.2}$ |
| *# best-performing tasks* | **17** | 4 | 5 |
| performance profile $\rho(5)$ | **1.0** | 0.35 | 0.5 |

### D.8. Study of ZOPO with More Prompt Candidates

Intuitively, generating more prompt candidates offers a closer approximation to the true function landscape. As our optimization method ZOPO is operated under a given set of prompt candidates, we here conduct an ablation study to examine the impact of a larger size of the generated prompt candidates (i.e., $|\mathcal{V}|$) on the optimization performance. For ZOPO, we use random soft prompts to feed Vicuna-13B and generate prompts until $\mathcal{V} = 500$ or $\mathcal{V} = 2000$. We compare the optimization results of ZOPO using the two different sizes of prompts, and the results are shown in Table 12. We also follow the APE generation template to prompt ChatGPT to generate different sizes of prompt candidates and use SBERT to produce their embeddings. For ChatGPT-generated prompts in ZOPO$_{\text{GPT}}$, we also consider two settings, $\mathcal{V} = 500$ or $\mathcal{V} = 1000$ (due to budget constraint). The corresponding result is shown in Table 13. We observe from the two tables that a larger set of prompt candidates may not necessarily lead to strictly better performance, and generating a relatively small set of strong prompt candidates (e.g., of size 500) is already good enough when we aim to find the optimal prompt.

Table 12: Ablation study of different sizes of prompt candidates in ZOPO.

| Tasks | $|\mathcal{V}| = 500$ | $|\mathcal{V}| = 2000$ |
|---|---|---|
| antonyms | $85.2_{\pm 3.2}$ | $\mathbf{86.3}_{\pm 0.9}$ |
| auto_categorization | $32.7_{\pm 1.9}$ | $\mathbf{37.3}_{\pm 1.2}$ |
| auto_debugging | $\mathbf{41.7}_{\pm 15.6}$ | $33.3_{\pm 11.8}$ |
| cause_and_effect | $\mathbf{94.7}_{\pm 3.7}$ | $\mathbf{94.7}_{\pm 1.9}$ |
| common_concept | $\mathbf{23.5}_{\pm 3.4}$ | $17.0_{\pm 6.1}$ |
| diff | $\mathbf{100.0}_{\pm 0.0}$ | $\mathbf{100.0}_{\pm 0.0}$ |
| informal_to_formal | $\mathbf{61.3}_{\pm 2.7}$ | $56.6_{\pm 4.1}$ |
| letters_list | $\mathbf{100.0}_{\pm 0.0}$ | $\mathbf{100.0}_{\pm 0.0}$ |
| negation | $\mathbf{86.3}_{\pm 0.5}$ | $\mathbf{86.3}_{\pm 0.5}$ |
| object_counting | $52.3_{\pm 6.6}$ | $\mathbf{53.0}_{\pm 6.5}$ |
| odd_one_out | $\mathbf{32.0}_{\pm 11.3}$ | $20.7_{\pm 6.6}$ |
| orthography_starts_with | $\mathbf{56.5}_{\pm 12.6}$ | $46.0_{\pm 6.9}$ |
| rhymes | $\mathbf{100.0}_{\pm 0.0}$ | $\mathbf{100.0}_{\pm 0.0}$ |
| second_word_letter | $25.7_{\pm 4.7}$ | $\mathbf{35.3}_{\pm 27.5}$ |
| sentence_similarity | $7.6_{\pm 9.3}$ | $\mathbf{24.7}_{\pm 6.1}$ |
| sum | $\mathbf{100.0}_{\pm 0.0}$ | $\mathbf{100.0}_{\pm 0.0}$ |
| synonyms | $\mathbf{43.3}_{\pm 0.9}$ | $40.0_{\pm 3.3}$ |
| taxonomy_animal | $90.0_{\pm 7.1}$ | $\mathbf{91.3}_{\pm 7.6}$ |
| word_sorting | $\mathbf{60.0}_{\pm 4.2}$ | $59.0_{\pm 6.4}$ |
| word_unscrambling | $\mathbf{59.3}_{\pm 2.8}$ | $54.7_{\pm 3.3}$ |
| *# best-performing tasks* | **14** | 12 |
| performance profile $\rho(5)$ | **0.9** | 0.8 |

Table 13: Ablation study of different sizes of prompt candidates in ZOPO$_{\text{GPT}}$.

| Tasks | $|\mathcal{V}| = 500$ | $|\mathcal{V}| = 1000$ |
|---|---|---|
| antonyms | $\mathbf{84.0}_{\pm 1.4}$ | $80.3_{\pm 1.2}$ |
| auto_categorization | $27.0_{\pm 5.0}$ | $\mathbf{28.3}_{\pm 2.4}$ |
| auto_debugging | $29.2_{\pm 5.9}$ | $\mathbf{37.5}_{\pm 10.2}$ |
| cause_and_effect | $\mathbf{80.0}_{\pm 14.2}$ | $78.7_{\pm 3.8}$ |
| common_concept | $2.8_{\pm 0.6}$ | $\mathbf{11.7}_{\pm 6.8}$ |
| diff | $\mathbf{100.0}_{\pm 0.0}$ | $\mathbf{100.0}_{\pm 0.0}$ |
| informal_to_formal | $\mathbf{61.9}_{\pm 2.9}$ | $57.2_{\pm 8.9}$ |
| letters_list | $\mathbf{100.0}_{\pm 0.0}$ | $99.3_{\pm 0.5}$ |
| negation | $\mathbf{77.7}_{\pm 2.6}$ | $75.0_{\pm 1.6}$ |
| object_counting | $40.3_{\pm 0.5}$ | $\mathbf{41.3}_{\pm 1.2}$ |
| odd_one_out | $68.7_{\pm 2.5}$ | $\mathbf{72.0}_{\pm 0.0}$ |
| orthography_starts_with | $71.0_{\pm 0.0}$ | $\mathbf{71.3}_{\pm 0.9}$ |
| rhymes | $61.0_{\pm 2.8}$ | $\mathbf{100.0}_{\pm 0.0}$ |
| second_word_letter | $96.7_{\pm 2.4}$ | $\mathbf{99.7}_{\pm 0.5}$ |
| sentence_similarity | $\mathbf{37.3}_{\pm 0.9}$ | $0.0_{\pm 0.0}$ |
| sum | $\mathbf{100.0}_{\pm 0.0}$ | $\mathbf{100.0}_{\pm 0.0}$ |
| synonyms | $44.7_{\pm 4.1}$ | $\mathbf{45.3}_{\pm 1.7}$ |
| taxonomy_animal | $\mathbf{92.3}_{\pm 0.5}$ | $89.3_{\pm 1.9}$ |
| word_sorting | $\mathbf{60.3}_{\pm 3.1}$ | $54.3_{\pm 7.0}$ |
| word_unscrambling | $58.3_{\pm 1.9}$ | $\mathbf{60.3}_{\pm 2.5}$ |
| *# best-performing tasks* | 10 | **12** |
| performance profile $\rho(5)$ | 0.85 | **0.9** |

## D.9. Best Prompts Found

We list the best prompts discovered by our method ZOPO for every instruction induction task here in Table 14, which corresponds to the results in Table 3.

Table 14: The best prompts discovered by our method ZOPO for every instruction induction task, where "*" indicates the best prompt is found by ZOPO$_{\text{GPT}}$ for that task.

| Task | Best prompt |
|---|---|
| active_to_passive | The prompt was to convert the given sentence into passive voice. |
| antonyms | The prompt was to rewrite the given words into their opposite meaning. So, "humorless" becomes "humorous", "depressing" becomes "cheerful", "unwrap" becomes "wrap", "consumptive" becomes "generative", "uncoil" becomes "coil". |
| auto_categorization | The prompt was to input the given names and output the corresponding apparel. For example, the input "Nature Nanotechnology, Annual Review of Biochemistry, and The Lancet Neurology" would output as "top journals". |
| auto_debugging | The prompt was to write a program that would take the given input and output the expected output. For example, the first input was a simple calculation, and the expected output was "2550". The second input was a class definition with a method, and the expected output was "5". |
| cause_and_effect | The prompt was to identify the sentence that is the cause and the sentence that is the effect in each pair of sentences. The input sentences are given, and the output is the cause sentence. |
| common_concept | The prompt was to create a series of pairs of inputs and outputs, where the outputs are related to the inputs in some way. For example, the inputs "guitars" and "pendulums" are related to the output of "involve oscillations. |
| diff | The prompt was to subtract the second number from the first number. For example, the first input would be 41 and the second input would be 13, so the output would be 28 (41 - 13). The same process would be applied for the other inputs and outputs. |
| first_word_letter | The prompt was to create a program that takes a single input (a word representing a legal concept or term) and outputs a corresponding letter of the alphabet that represents that concept or term. For example, if the input is "year", the program should output "y". |
| informal_to_formal* | The prompt was to rephrase each input sentence using a more formal or polite language. |
| larger_animal | The prompt was to create a program that takes two input animals and outputs the animal that is bigger. The program uses the "¿=" operator to compare the size of the first animal to the size of the second animal. If the first animal is bigger, the program outputs the first animal. |
| letters_list | The prompt was to create a program that takes a single word input (e.g. "year") and outputs a concatenated string of letters and spaces that approximates the pronunciation of that word (e.g. "y e a r"). |
| negation | The prompt was to flip the truth value of the input statements. For example, if the input statement is "Cany Ash and Robert Sakula are both Architects," the output should be "Cany Ash and Robert Sakula are not Architects. |
| num_to_verbal | The prompt was to write a program that takes a number as input and outputs the number in words, using the appropriate number formatting. The examples provided in the input show the expected output for each number. |
| object_counting | The prompts were to provide the output of a given input, where the input is a list of items and the output is a number representing the total count of those items. The examples given in the prompt show how the prompts should be used to generate the desired output. |
| odd_one_out* | The prompt was to identify the word that is most different from the others in the group. |

| | |
|---|---|
| orthography_starts_with* | The prompt was to identify the first word that begins with a specific letter in each sentence. |
| periodic_elements | The prompts were to write a program that takes an input value and outputs the corresponding element name based on that value. For example, if the input is 24, the program would output "chromium. |
| rhymes | The prompts were to create a program that takes in a word as input and outputs a related word based on a specific set of rules. The rules are as follows: If the input word starts with "tri", the output should be "slip". |
| second_word_letter* | The prompt was to "Identify and return the second letter of the input word". |
| sentence_similarity* | The prompt was to create two different sentences that have similar meanings but are not identical. The output of each input-output pair indicates how closely the two sentences match in terms of meaning.

Explanation of outputs:
- 5 - perfectly: The two sentences are very similar in meaning and can be considered as equivalent.
- 3 - probably: The two sentences have some similarities in meaning but there are also some differences, making it less certain that they are equivalent.
- 2 - possibly: The two sentences have some similarities but also significant differences, making it unlikely that they are equivalent.
- 1 - probably not: The two sentences have very different meanings and are unlikely to be considered as equivalent.
- 0 - definitely not: The two sentences have no similarity in meaning and cannot be considered as equivalent. |
| sentiment | The prompt was to classify the given reviews as positive or negative based on the given input and output. The output is positive when the review is positive, and negative when the review is negative. |
| singular_to_plural | The prompt was to convert the input words to their plural form by adding "s" to the end of the word. This was done by using the "replace" function in Excel, which allows you to replace a specific text string with another text string. |
| sum | The prompt was to write a program that takes two numbers as input and outputs their sum as the result. The program uses the 'scanf' function to read the input numbers from the user, and the 'printf' function to display the result. |
| synonyms* | The prompt was to create a list of words that are synonyms or closely related to the given word. |
| taxonomy_animal* | The prompt was to select all the animals in the input and output them in the order they appear. |
| translation_en-de | The prompts were to input various words and have the model generate the corresponding output in German. It appears that the model was successful in generating the desired output for each of the input words provided. If there are any additional prompts or clarification needed, please let me know. |
| translation_en-es | The prompts were to translate a set of words from Spanish to English using the provided translation table. |
| translation_en-fr | The prompt was to input a word and then output the corresponding word in French. It appears that the input and output words are being matched correctly, with the exception of the word "initiative," which should have the output "initiative" in French, not "enterprise. |
| word_sorting* | The prompt was to alphabetize the input list in ascending order and provide the resulting output as a list. |
| word_unscrambling | The prompt was to create a program that takes an input word and outputs the corresponding word with the letters rearranged in order. For example, given the input "eccpat", the program should output "accept". |

