# OpenReview forum: "Localized Zeroth-Order Prompt Optimization"
_ICML.cc/2024/Workshop/ICL — ICML 2024 Workshop ICL Poster_

### Official Review · Reviewer_EhoS · 2024-06-01
**Well-motivated methodology, lacks extensive experimentation and results are not particularly convincing**

**Rating:** 2
**Fit:** 2
**Confidence:** 2

**Workshop Review:**

This paper demonstrates the intuitions that (1) local optimums in prompt optimization are common and perform fairly well and (2) the input domain can significantly affect the performance to find optimums. The authors then propose a new method called ZOPO that they claim outperforms prior baseline methods of prompt optimization.

In general, the paper is well motivated and demonstrates interesting insights. However, the proposed methodology is held back by unconvincing experiments that do not seem to show that ZOPO necessarily outperforms baselines. Furthermore, there seems to be a lack of thorough testing with many model families and across model scale, which puts into question the generalizability of the work.

**Reason For Not Giving Higher Score:**

There also exist some grammatical/syntax/stylistic errors. For example, the sentence "In all, we conclude our aforementioned insights as below." is duplicated in Section 3.2.

I may have missed it but it wasn't immediately clear to me how the "query budget of 165" was chosen in the beginning of Section 5.

It's not super clear to me that ZOPO necessarily achieves superior performance. Figure 5 seems to show that other methods (InstructZero on `informal_to_formal` and OPRO on `taxonomy_animal` and `cause_and_effect`) are able to match or even outperform ZOPO. Additionally, Table 1 seems to show that standard ZOPO does not significantly outperform baselines on many tasks, and that ZOPO-GPT is the version of ZOPO that is better (although I'm not confident if it's a fair comparison because the baseline method numbers are presumably retrieved using a weaker model than ChatGPT).

In general, experiments seem to have been done using Vicuna-13B and ChatGPT, so it's not clear how well the results generalize to other models. I would have been interested to see whether the results hold with respect to e.g., model scale, as prior work has shown that ICL abilities can vary as scale increases [1].

[1] Jerry Wei, Jason Wei, Yi Tay, Dustin Tran, Albert Webson, Yifeng Lu, Xinyun Chen, Hanxiao Liu, Da Huang, Denny Zhou, and Tengyu Ma. Larger language models do in-context learning differently. 2023. https://arxiv.org/abs/2303.03846.

**Reason For Not Giving Lower Score:**

The paper still shows some interesting analyses of in-context learning and is well-motivated. In particular, Figure 2 shows interesting visualization of the intuition that we need not necessarily optimize for a global optimum but rather that local optimums get most of the way there. Moreover, Figure 3 demonstrates the importance of input domains in affecting the performance of in-context learning, which is often overlooked by other works that simply focus on improving performance in particular domains.

---

### Official Review · Reviewer_azC5 · 2024-06-10
**Interesting idea**

**Rating:** 2
**Fit:** 3
**Confidence:** 2

**Workshop Review:**

This paper takes on the problem of prompt optimization. They identify a number of "local minima" but actually it is not straightforward to verify from Figure 2 that there are local minima. Then, they demonstrate that the choice of initialization matters a lot for which local optimum is found. These two observations together suggest that prompt optimization (unsurprisingly) operates in the kernel regime. The authors then apply a previously existing query-efficient zeroth-order optimization technique to optimize the prompt. The algorithm permits them to plug in a kernel function, for which they use the neural tangent kernel of the model. The resulting algorithm does well on many instruction induction tasks.

**Reason For Not Giving Higher Score:**

The primary weakness in this algorithm is that it requires time quadratic in the number of examples to compute the kernel. Each entry in this kernel matrix requires differentiating the given model, so it is quite expensive. I'm unfamiliar with the baselines and don't know how much the cost of ZOPO is relative to those, but I imagine it's higher because ZO algorithms are query-inefficient and the kernel computation is expensive. Authors should also compare to something like prefix tuning, where prompts are continuous, if they are considering the setting where the model is differentiable anyways. I recommend the authors add a citation to this relevant work on fine-tuning for GLUE tasks being in the kernel regime: https://openreview.net/forum?id=49dTFIGdx8

**Reason For Not Giving Lower Score:**

Work has extensive experiments and is a sound argument.

---

### Meta-Review · Area_Chair_PjGy · 2024-06-14

**Recommendation:** 2

**Metareview:**

This paper proposes a novel prompt optimization method called Localized zeroth-order prompt optimization (ZOPO) which incorporates a Neural Tangent Kernel-derived Gaussian process into standard zeroth-order optimization. ZOPO is motivated by the intuitions that local optima in prompt optimization are common and that the input domain can significantly affect performance.

All reviewers have favorable opinion of this paper, noting good motivation, but expressing concerns about the experimental results.

---

### Decision · Program_Chairs · 2024-06-17

Accept (Poster)